# Active learning of neural population dynamics using two-photon holographic optogenetics

**Andrew Wagenmaker***
University of California, Berkeley

**Lu Mi***
Georgia Tech

**Marton Rozsa**
Allen Institute for Neural Dynamics

**Matthew S. Bull**
Allen Institute for Brain Science

**Karel Svoboda**
Allen Institute for Neural Dynamics

**Kayvon Daie**†
Allen Institute for Neural Dynamics

**Matthew D. Golub**†
University of Washington

**Kevin Jamieson**†
University of Washington

## Abstract

Recent advances in techniques for monitoring and perturbing neural populations have greatly enhanced our ability to study circuits in the brain. In particular, two-photon holographic optogenetics now enables precise photostimulation of experimenter-specified groups of individual neurons, while simultaneous two-photon calcium imaging enables the measurement of ongoing and induced activity across the neural population. Despite the enormous space of potential photostimulation patterns and the time-consuming nature of photostimulation experiments, very little algorithmic work has been done to determine the most effective photostimulation patterns for identifying the neural population dynamics. Here, we develop methods to efficiently select which neurons to stimulate such that the resulting neural responses will best inform a dynamical model of the neural population activity. Using neural population responses to photostimulation in mouse motor cortex, we demonstrate the efficacy of a low-rank linear dynamical systems model, and develop an active learning procedure which takes advantage of low-rank structure to determine informative photostimulation patterns. We demonstrate our approach on both real and synthetic data, obtaining in some cases as much as a two-fold reduction in the amount of data required to reach a given predictive power. Our active stimulation design method is based on a novel active learning procedure for low-rank regression, which may be of independent interest.

## 1 Introduction

Neural population dynamics describe how the activities across a population of neurons evolve over time due to local recurrent connectivity and inputs to the population from other neurons or brain areas. Identifying these population dynamics can provide critical insight into the computations performed by a neural population [1]. Dynamical systems models have enabled neuroscientists to generate and test a multitude of hypotheses about how specific neural populations support the neural computations that underlie, for example, motor control [2–4], motor timing [5, 6], decision making [7–10], working memory [11], social behavior [12], and learning [13–16].

---

*Equally contributing first authors.
†Equally contributing senior authors.
Correspondence to: `ajwagen@berkeley.edu` or `lmi7@gatech.edu`.

38th Conference on Neural Information Processing Systems (NeurIPS 2024).

The traditional approach to data-driven modeling of a neural population typically involves two separate stages. First, neural population activity is recorded while an animal performs a task of interest. Then, a dynamical systems model is fit to the recorded neural responses [17–33]. This approach suffers from two key limitations. First, any inferred structure is purely correlational, and cannot be interpreted with any notion of causality. Second, the experimenter has limited control over how the neural population dynamics are sampled, which can lead to inefficient data collection—oversampling in some parts of neural activity space while altogether missing others. Given constraints on time and resources in neurophysiological experiments, there is a strong need for techniques that minimize the amount of experimental data required to identify the neural population dynamics.

We seek to overcome these limitations by actively designing the causal circuit perturbations that will be most informative to learning a dynamical model of the neural population response. For circuit perturbations, we employ two-photon holographic photostimulation (Figure 1), which provides temporally precise, cellular-resolution optogenetic control over the activity of ensembles of neurons [34–41]. When paired with two-photon calcium imaging, photostimulation protocols can provide insight into network connectivity by enabling the measurement of the causal influence that each perturbed neuron exerts on all other recorded neurons [36, 39, 42–46]. This platform enables targeted excitation of the neural population dynamics, thus providing the experimenter with unprecedented control over the data collected for informing a model of the neural population dynamics.

Here, we develop active learning techniques for designing photostimulation patterns that allow for efficient estimation of low-rank neural population dynamics and the underlying network connectivity. First, we introduce a low-rank autoregressive model that captures low-dimensional structure in neural population dynamics and allows inference of the causal interactions between recorded neurons. We then propose an active learning procedure which chooses photostimulations to target this low-dimensional structure, and demonstrate it in two settings: estimating the underlying causal interactions when using the learned autoregressive model as a simulator of the true dynamics, and adaptively selecting which samples to observe from our dataset of neural population activity recorded via two-photon calcium imaging of mouse motor cortex in response to two-photon holographic photostimulation. In both cases, we show that our active approach obtains substantially more accurate estimates with fewer measurements compared to passive baselines. Our methodology is based on a novel analysis of nuclear-norm regression with non-isotropic inputs. To the best of our knowledge, this is the first approach to demonstrate significant gains applying active learning to low-rank matrix estimation problems, and thus we believe this may be of independent interest.

## 2 Related Work

**Modeling Neural Responses to Stimulation.** Many studies have applied direct electrical or optical stimulation to neural populations to probe the dynamical properties of neural circuits and their relation to circuit function [4, 10, 26, 47–49]. However, these stimulation techniques lack the spatial specificity needed to precisely probe the causal influence of individuals neurons on the population dynamics, and these experimental designs were *passive* in that the stimulation protocols were specified prior to data collection (with [44, 48] as notable exceptions). Other work has explored a related but separate problem of minimizing off-target effects when photostimulating individual neurons [41].

**Low-Rank Matrix Recovery.** Low-rank matrix recovery has been intensively researched over the last decade and a half [50–52]. However, existing analyses rely critically on the assumption that the set of measurements taken are highly symmetric and satisfy some notion of the restricted isometry property (RIP) or incoherence. The matrix recovery problem of our setting departs from the classical literature in several ways. First, the set of feasible measurements we can take is constrained by the physical limits of the photostimulation system. Second, as we aim to *adapt* and *actively* learn these matrix coefficients, we should expect that our resulting set of measurements should be highly skewed by design. Motivated by this, we develop, to the best of our knowledge, the first bounds on low-rank estimation using the nuclear norm heuristic that gives a quantification of the estimation error in terms of the precise individual measurements taken (i.e., in contrast to a more global property like RIP).

**Active Learning and Low-Rank Estimation.** The active learning literature is vast, and a full survey is beyond the scope of this work. We focus in particular on active learning for dynamical systems, and problems with low-rank structure. The estimation of dynamical systems—the *system identification* problem—is central to many areas of engineering and science [53]. The problem of actively designing

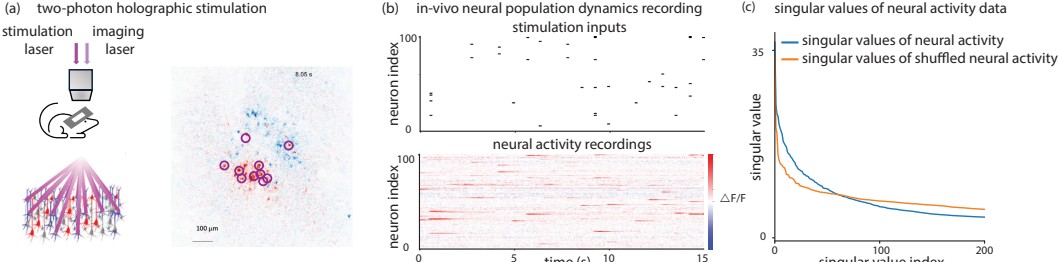

Figure 1: (a) Two-photon imaging and holographic photostimulation platform (left) and a representative image frame (right). Purple circles indicate neurons photostimulated immediately before frame acquisition. Red and blue indicate increases and decreases of firing activity, respectively, relative to before photostimulation. (b) Example time series photostimulation inputs (top) and neural responses (bottom) from 100 randomly selected neurons (out of $d = 663$ recorded neurons identified in the FoV). (c) Neural responses $y_t$ occupy a low-dimensional subspace. Singular values from a representative dataset's demeaned neural activity data matrix (blue) indicate substantially more data variance residing in a few dozen dimensions (out of the full $d = 663$ dimensional neural activity space) than is expected by chance (orange, singular values when removing low-dimensional structure by shuffling time indices independently for each neuron; note clipped horizontal axis).

inputs to effectively estimate the parameters of a dynamical system has been studied extensively for decades [54–61]. More recently, a variety of provably efficient approaches have been developed for both linear [62, 63] and nonlinear [64, 65] systems. Other related work has considered active learning for latent variable models [66], which are often effective models of neural dynamics. As compared to these works, a key feature of our setting is the low-rank structure present in the data, which to our knowledge has not been previously studied within the active system identification literature.

Beyond dynamical systems, some attention has been devoted to active learning with low-rank structure, in particular works on low-rank bandits [67–70]. While the setting considered in these works is somewhat different—they aim to solve a bandit problem, while we are interested in regression—they similarly seek to develop active learning approaches which make efficient use of low-rank structure. Also related is the work of [71], which shows that in the related sparse estimation setting, there does not exist more than a logarithmic gain to being adaptive. The results of this work are minimax, however—only applying to certain "hard" problems—and do not address the matrix recovery problem.

## 3 Preliminaries

**Dataset Details.** Neural population activity was recorded in mouse motor cortex using two-photon calcium imaging at 20Hz of a 1mm×1mm field of view (FoV) containing 500-700 neurons. Each recording spanned approximately 25 minutes and 2000 photostimulation trials. In each trial, a 150ms photostimulus was delivered and was followed by a 600ms response period before the next trial began. Each photostimulus targeted a group of 10-20 randomly selected neurons, and a total of 100 unique photostimulation groups were defined for each experiment ($\approx$ 20 trials per group). We evaluate our techniques on four such datasets.

### 3.1 Fitting Low-Rank Dynamical Models

We first seek to develop effective dynamical models of the neural activity in our photostimulation datasets. Obtaining such models will provide insight into which photostimuli are most informative, and gives us a means to evaluate the effectiveness of our active learning methods. We consider three classes of models: autoregressive (AR) models, low-rank AR models, and nonlinear RNN models. Results from fitting these models are shown in Figure 2. We describe the model details next.

At discrete time $t \in \mathbb{N}$, we denote the true neural activity across the $d$ imaged neurons as $x_t \in \mathbb{R}^d$, the noisy, measured activity as $y_t \in \mathbb{R}^d$, and the photostimulus intensity applied across those same $d$ neurons as $u_t \in \mathbb{R}^d$. Applying stimulus $u_t$ influences the measured neural activity at the next timestep $y_{t+1}$. However, just the snapshot $y_t$ may not capture the full true *state* of the neural population, which may include not just the current neural activity, but potentially also multiple orders of temporal derivatives. To capture these effects, we consider an AR-$k$ model defined as:

$$x_{t+1} = \sum_{s=0}^{k-1}(A_s x_{t-s} + B_s u_{t-s}) + v, \quad y_t = x_t + w_t \quad \text{with} \quad w_t \sim \mathcal{N}(0, \sigma^2 I_d), \quad (3.1)$$

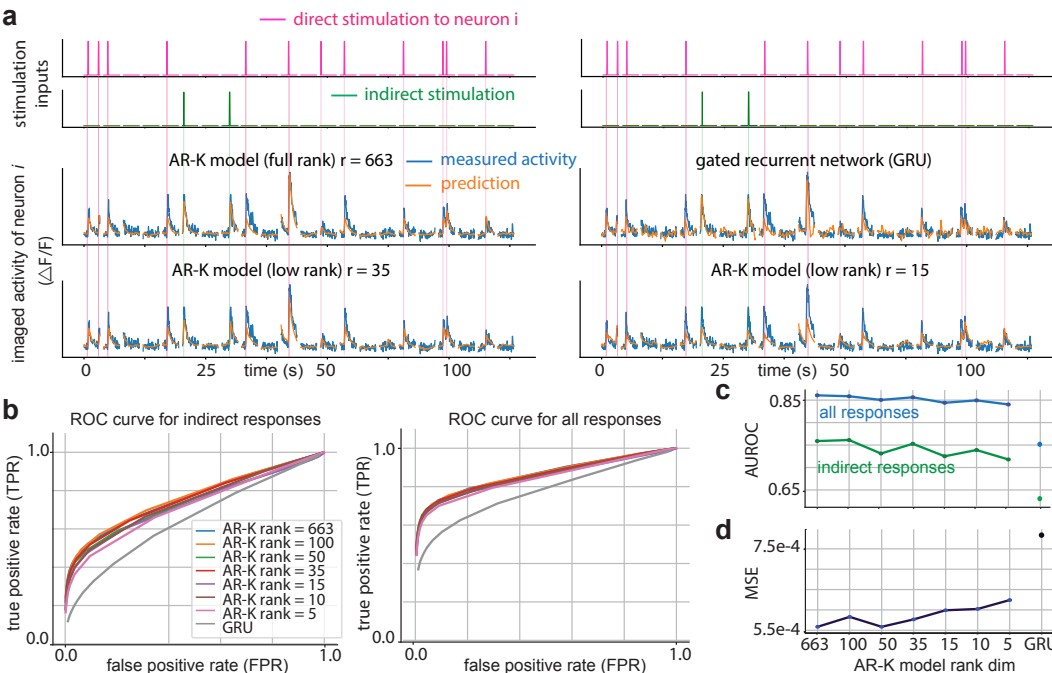

Figure 2: Example data and cross-validated model predictions. (a) Roll-out predictions of the activity of an example neuron $i$ using low-rank AR-$k$ models ($k = 4$) and GRU networks for 22 example data segments (3.3s per segment; segments separated by brief horizontal spaces). Each model's predictions are seeded with the first $k = 4$ timesteps (200ms) of activity from $d = 663$ neurons and are then unrolled to predict the activity across all $d$ neurons over the next 66 timesteps, given the full 70-timestep sequence of photostimulation to all $d$ neurons. Most responses of neuron $i$ are tied to "direct" photostimulation of neuron $i$ (pink, first row of panels). Several "indirect responses" are tied to stimulation of other neurons $j \neq i$ that influence neuron $i$ through the population dynamics. To avoid showing all indirect stimuli (to $d - 1$ neurons), only select indirect stimuli are shown (green, second row of panels). (b) Receiver operator characteristic (ROC) curve of true-positive rate and false-positive rate for response detection are calculated on indirect responses only (left) and all direct and indirect responses (right). (c) Area under ROC curve (AUROC) and (d) mean square error (MSE) for all predictions.

where $A_s \in \mathbb{R}^{d \times d}$ and $B_s \in \mathbb{R}^{d \times d}$ describe the coupling between neurons and stimulus at the time lag of $s$ timesteps, $s = 0, \ldots, k - 1$, and offset $v \in \mathbb{R}^d$ accounts for baseline neural activity. Given input-observation pairs $\{(u_t, y_t)\}_t$, the coefficients $\{(A_s, B_s)_{s=0}^{k-1}, v\}$ of (3.1) can be fit using least squares. Despite its simplicity, this linear model reproduces the recorded neural activity remarkably well (see "full rank" model of Figure 2).

Neural population dynamics are frequently reported as residing in a subspace of lower dimension than the total number of recorded neurons [2, 7, 17, 72–77]. The population dynamics in our datasets are consistent with such low-dimensional structure, as indicated by the singular value spectrum in Figure 1(c). Inspired by this observation, we introduce a set of low-rank dynamical models, where each matrix of $\{(A_s, B_s)_{s=0}^{k-1}\}$ is re-defined as diagonal plus low-rank. Explicitly, we parameterize $A_s = D_{A_s} + U_{A_s} V_{A_s}^\top$ and $B_s = D_{B_s} + U_{B_s} V_{B_s}^\top$, where $D \in \mathbb{R}^{d \times d}$ with $D_{ij} = 0$ for all $i \neq j$, $U \in \mathbb{R}^{d \times r}$, and $V \in \mathbb{R}^{d \times r}$ for predefined rank $r$. The diagonal matrices account for substantial autocorrelation in each neuron's activity ($D_{A_s}$) and for the reliable response of each neuron to direct photostimulation ($D_{B_s}$), whereas the low-rank matrices ($UV^\top$) confer coupling between neurons. To fit these parameters, we optimize the following objective function with gradient descent over all parameters:

$$\underset{A_s, B_s \in \mathbb{R}^{d \times d}, s=0, \ldots, k-1, v \in \mathbb{R}^d}{\text{minimize}} \sum_{t=1}^{T} \left( y_{t+1} - \sum_{s=0}^{k-1} A_s y_{t-s} - \sum_{s=0}^{k-1} B_s u_{t-s} - v \right)^2. \qquad (3.2)$$

Figure 2 shows that these low-rank models perform comparably to the full rank versions in terms of predictive performance; indeed the rank $r = 35$ model appears almost indistinguishable from the full

rank model. From a statistical perspective, low-rank models have far fewer degrees of freedom, and hence require less data to fit.

To assess whether more expressive nonlinear models could be advantageous, we also fit a gated recurrent unit (GRU) network model, adapted from [22], as shown in Figure 2. Interestingly, the GRU model did not perform as well as the AR-$k$ models, potentially due to the complexities of hyperparameter tuning. Therefore, we focus on linear models in the analysis that follows. Additional details on model fitting are provided in Appendix B.2.

## 3.2 The Causal Connectivity Matrix

While we require dynamical models to predict the temporal evolution of the neural population activity, we are also interested in inferring how the activity of one recorded neuron causally influences the activity of the other recorded neurons. To address this need, we define a *causal connectivity matrix*, $H \in \mathbb{R}^{d \times d}$, to be the mapping such that $Hu \in \mathbb{R}^d$ quantifies the total response (across time) of each neuron to a single-timestep photostimulus $u$. That is, $\sum_{t=1}^{\infty} x_t = Hu$, where $x_1, x_2, x_3, \ldots$ are the neural activities generated by the population dynamics if $u_0 = u, u_{t \geq 1} = 0$, and $x_{t \leq 0} = x_\infty$ is the steady state or resting state of the system subject to no photostimulation. If the dynamics are linear, or more specifically follow (3.1), such a matrix $H$ is guaranteed to exist, and can be formed by simply rolling out (3.1) with the appropriate initializations. While $H$ is not explicitly constrained to be low-rank, if it is obtained from a low-rank AR-$k$ model, it too will exhibit low-rank structure.

In our experimental paradigm, photostimulation acts as a causal perturbation to the population dynamics, and as such, our statistical framework is able to capture causal interactions, as opposed to merely correlative interactions. This is in contrast to the majority of work on neural population dynamics, which involves fitting dynamical models to passively obtained data. Due to the lack of causal manipulations in these studies, one cannot distinguish whether statistical relationships arise between neurons due to correlation (e.g., due to a shared upstream influence) versus causation (e.g., neuron $i$ directly influences neuron $j$). Such correlative relationships are typically referred to as "functional connectivity"; we instead use the term "causal connectivity" to convey the additional causal interpretability afforded in our setting.

To fit $H$, we could first fit $\{\widehat{A}_s, \widehat{B}_s\}_{s=0}^{k-1}$ and then use these as plug-in estimates for their true values to compute $H$. Alternatively, we take a more direct approach inspired by the definition of $H$ itself. By inspecting the raw data of Figure 2(a) and observing the rate at which each stimulated neuron returns to baseline activity, it is clear that the system mixes (i.e., forgets the past) quickly. This suggests that the total response due to input $u$ asymptotes after some finite number of timesteps $\tau$. Thus, we can apply some photostimulus $u \in \mathbb{R}^d$ at time $t = 0$ and then measure the total response $z = \sum_{t=1}^{\tau} y_t$, where $y_t \in \mathbb{R}^d$ is the noisy measurement of the true neural response $x_t$. If we repeat this for many pairs $\{(u_n, z_n)\}_n$ then we can approximate $H$ as

$$\widehat{H} := \arg\min_{H'} \sum_n \|z_n - H'u_n\|_2^2.$$

In this work we adopt this latter approach. Since we believe $H$ to be low rank, this amounts to a low-rank matrix recovery problem with matrix-vector observations. In the next section, we will describe how to adaptively choose $\{u_n\}_n$ to estimate $H$ using as few (stimulus, response) pairs as possible. Subsequently in Section 5, we will demonstrate that actively designing inputs to accelerate the learning of $H$ effectively accelerates the learning of the full dynamics as well.

## 4 Active Learning of Low-Rank Matrices

In the previous section, we saw that estimating the causal connectivity matrix $H$ induced by the neural population dynamics amounts to low-rank matrix recovery, where we apply some photostimulus $u \in \mathbb{R}^d$ and observe the neural population response $z \approx Hu$ plus noise. In this section we seek to understand how we should choose the photostimuli to estimate the causal connectivity as quickly as possible. To this end, in Section 4.1 we present novel results characterizing the estimation error of the nuclear norm regression estimator, and in Section 4.2 present an algorithm motivated by these results which seeks to actively estimate low-rank matrices. These results will directly motivate a procedure for designing photostimulation inputs.

To demonstrate the generality of our results, in Section 4.1 we consider a general matrix regression setting. In particular, let $\Theta_\star \in \mathbb{R}^{d_1 \times d_2}$ be a rank $r$ (potentially non-square) matrix, $\varphi_n \in \mathbb{R}^{d_1 \times d_2}$

some input matrix, and assume scalar observations:

$$z_n = \langle \Theta_\star, \varphi_n \rangle + \eta_n, \quad \eta_n \sim \mathcal{N}(0,1), \tag{4.1}$$

where $\langle \Theta_\star, \varphi_n \rangle = \mathrm{tr}(\Theta_\star^\top \varphi_n)$ for $\mathrm{tr}(\cdot)$ the trace of a matrix. Note that the setting considered in Section 3.2 is a special case of this observation model with $\Theta_\star \leftarrow H$ and, for each input stimulation $u$, measuring the response of (4.1) to $d$ inputs $\varphi_j$ of the form $\varphi_j \equiv \mathbf{e}_j u^\top$ for $j = 1, \ldots, d$.

**Matrix Notation.** We let $\| \cdot \|_{\mathrm{F}}, \| \cdot \|_{\mathrm{op}}, \| \cdot \|_*$ denote the Frobenius, operator, and nuclear norm of a matrix, respectively. $\dagger$ denotes the pseudo-inverse of a matrix. $\mathrm{vec}(\cdot)$ denotes the vectorization of a matrix, and $\mathrm{mat}(\cdot)$ the inverse of the vectorization. We also let $\triangle_\mathcal{U}$ denote the simplex—the set of distributions—over a set $\mathcal{U}$.

## 4.1 Constrained Nuclear Norm Estimator under Non-Isotropic Measurements

We are interested in understanding how we can effectively take into account the low-rank structure of $\Theta_\star$, if our goal is to estimate $\Theta_\star$ from the observations of (4.1). To this end, we consider the following nuclear-norm constrained least-squares estimator for $\Theta_\star$:

$$\widehat{\Theta} = \arg\min_{\Theta \in \mathcal{K}} \|\Phi(\Theta) - z\|_2^2 := \textstyle\sum_{n=1}^N (\langle \varphi_n, \Theta \rangle - z_n)^2 \quad \text{for} \quad \mathcal{K} := \{\Theta : \|\Theta\|_* \leq \|\Theta_\star\|_*\}, \tag{4.2}$$

where here we let $\Phi(\Theta_\star) \in \mathbb{R}^N$ denote the vector where the $n$th element is $\langle \varphi_n, \Theta_\star \rangle$, and $z = \Phi(\Theta_\star) + \eta$ the vector of observations, for $\eta$ the vector with elements $\eta_n$. Define $\Theta_\star = U\Sigma V^\top$ as the skinny SVD such that $U \in \mathbb{R}^{d_1 \times r}$, $V \in \mathbb{R}^{d_2 \times r}$, and consider the linear projection operators $P_\perp, P_\parallel : \mathbb{R}^{d_1 \times d_2} \to \mathbb{R}^{d_1 \times d_2}$ defined as:

$$P_\perp(M) := (I - UU^\top)M(I - VV^\top) \quad \text{and} \quad P_\parallel(M) := M - P_\perp(M),$$

for any $M \in \mathbb{R}^{d_1 \times d_2}$. We call $P_\parallel$ the projection onto the *tangent space* of $\Theta_\star$. Note that the dimension of the range of $P_\parallel$ is equal to just $r(d_1 + d_2) - r^2 \ll d_1 d_2$. We are now ready to state our main result on the estimation error of $\widehat{\Theta}$, for $\widehat{\Theta}$ as defined in (4.2).

**Theorem 1.** *Define* $\mu := \|(\Phi^*\Phi)^{1/2}((P_\parallel \Phi^*\Phi P_\parallel)^\dagger)^{1/2}\|_{\mathrm{op}}$. *Then with probability at least* $1 - 2\delta$:

$$\|\widehat{\Theta} - \Theta_\star\|_{\mathrm{F}} \leq \frac{4}{1-\mu} \sqrt{\mathrm{tr}\big((P_\parallel \Phi^*\Phi P_\parallel)^\dagger\big) + 2\|(P_\parallel \Phi^*\Phi P_\parallel)^\dagger\|_{\mathrm{op}} \log \frac{d_1 d_2}{\delta}}$$

$$+ 4\|P_\perp(\Phi^*\Phi)^{1/2}\|_{\mathrm{op}} \|(P_\parallel \Phi^*\Phi P_\parallel)^\dagger\|_{\mathrm{op}} (\sqrt{d_1} + \sqrt{d_2} + \sqrt{2\log\tfrac{1}{\delta}}),$$

*where here* $\Phi^*\Phi(M) := \sum_n \varphi_n \langle \varphi_n, M \rangle$ *and* $\mathrm{tr}(\cdot)$ *describes the sum of the eigenvalues of the linear operator* $(P_\parallel \Phi^*\Phi P_\parallel)^\dagger : \mathbb{R}^{d_1 \times d_2} \to \mathbb{R}^{d_1 \times d_2}$.

Theorem 1 provides a precise bound on the estimation error of the nuclear norm estimator under arbitrary inputs $\{\varphi_n\}_n$. To the best of our knowledge, this is the first such characterization of this estimator. This characterization is particularly essential in active learning problems, such as the problem considered here, where it is critical that we understand precisely how the estimation error scales with different inputs, in order to determine which inputs will most effectively reduce the estimation error. As the observation model of Section 3.2 is a special case of the setting considered in (4.1) with $\Theta_\star \leftarrow H$, Theorem 1 provides a quantification of how quickly we can estimate the causal connectivity matrix given some set of inputs; we expand on the implications of this connection in Section 4.2.

Theorem 1 states that the estimation error of the estimator (4.2) scales (predominantly) with the strength of our inputs $\varphi_n$ in the tangent space of $\Theta_\star$. Indeed, if $[w_1, \ldots, w_{d_1}]$ and $[v_1, \ldots, v_{d_2}]$ are the left and right singular vectors of the full SVD of $\Theta_\star$, and $L \in \mathbb{R}^{d_1 d_2 \times r(d_1 + d_2) - r^2}$ is a matrix with orthonormal columns $\mathrm{vec}(w_i v_j^\top)$ for $(i, j) : \{i \leq r\} \cup \{j \leq r\}$, then

$$\mathrm{tr}\big((P_\parallel \Phi^*\Phi P_\parallel)^\dagger\big) = \mathrm{tr}\big((L^\top \textstyle\sum_{n=1}^N \mathrm{vec}(\varphi_n)\mathrm{vec}(\varphi_n)^\top L)^\dagger\big),$$

so we see that the estimation error depends only on the scaling of $\sum_{n=1}^N \mathrm{vec}(\varphi_n)\mathrm{vec}(\varphi_n)^\top$ in the space spanned by $\mathrm{vec}(u_i v_j^\top)$ for $i \leq r$ or $j \leq r$—the tangent space to $\Theta_\star$. As an example of how this scales, assume that for $n = 1, \ldots, N$ the entries of each $\varphi_n$ are IID $\mathcal{N}(0,1)$ and $N \geq r(d_1 + d_2) - r^2$.

Then $\mu \approx 0$, $\mathrm{tr}\big((P_\parallel \Phi^* \Phi P_\parallel)^\dagger\big) \approx \frac{r(d_1+d_2)-r^2}{N}$, $\|(P_\parallel \Phi^* \Phi P_\parallel)^\dagger\|_{\mathrm{op}} \approx \frac{1}{N}$, and $\|P_\perp (\Phi^* \Phi)^{1/2}\|_{\mathrm{op}} \approx \sqrt{N}$. This translates to a bound of $\|\widehat{\Theta} - \Theta_\star\|_{\mathrm{F}}^2 \leq \frac{r(d_1+d_2)-r^2+\log(1/\delta)}{N}$. Critically, we see that this does not scale with the total number of parameters, $d_1 d_2$, but instead with $r(d_1 + d_2)$, which could be much smaller. The following result, due to [78], provides a lower bound on the estimation error of any unbiased estimator, and shows that the rate obtained by Theorem 1 is essentially unimprovable.

**Theorem 2** (Corollary 1 of [78])**.** *For unbiased estimator* $\widehat{\Theta}$, $\mathbb{E}[\|\widehat{\Theta} - \Theta_\star\|_{\mathrm{F}}^2] \geq \mathrm{tr}\big((P_\parallel \Phi^* \Phi P_\parallel)^\dagger\big)$.

### 4.2 Active Learning for Low-Rank Matrix Estimation

Given the above characterization, we turn now to the active learning problem: how can we best choose our inputs $\varphi_n$ to speed up estimation error of $\Theta_\star$? For simplicity, rather than the general matrix regression setting of (4.1), we consider here the vector regression case, as this is the setting of interest in learning the causal connectivity. In particular, assume that we play some $u_n \in \mathbb{R}^{d_2}$ and observe $z_n = \Theta_\star u_n + \eta_n$, for $\eta_n \sim \mathcal{N}(0, I_{d_1})$. A single vector observation corresponds to observing $d_1$ observations from (4.1), the responses to the matrix inputs $\varphi_j \equiv e_j u_n^\top$ for $j = 1, \ldots, d_1$. Assume that $\Theta_\star$ is rank $r$ and let $V_0 := [v_1, \ldots, v_r]$ denote the first $r$ right singular vectors of the full SVD of $\Theta_\star$. Then we have that:

$$\mathrm{tr}\big((P_\parallel \Phi^* \Phi P_\parallel)^\dagger\big) = (d_1 - r) \cdot \mathrm{tr}\big((V_0^\top \Sigma_N V_0)^\dagger\big) + r \cdot \mathrm{tr}\big((\Sigma_N)^\dagger\big), \quad \Sigma_N := \sum_{n=1}^N u_n u_n^\top. \quad (4.3)$$

This calculation, combined with Theorem 1, shows that the estimation error of $\Theta_\star$ scales with a weighting of two terms: one quantifying the amount of input energy we put into directions spanned by the top-$r$ right singular vectors, and one that quantifies the amount of input energy played isotropically (that is, in all directions). Note, however, that the input energy played in directions $V_0$ is weighted by a factor of $d_1 - r \approx d_1$, much larger weight than the weight of $r$ given to the term quantifying the isotropic input energy. This suggests that, to minimize the estimation error of $\Theta_\star$, we should focus a large portion of our sampling budget to target the directions spanned by the top-$r$ right singular vectors of $\Theta_\star$.

This strategy admits a transparent intuition. If $\Theta_\star$ is rank-$r$ and some vector $u$ is orthogonal to the top-$r$ right singular vectors of $\Theta_\star$, then $\Theta_\star u = 0$. Thus, if we *know* what subspace the top-$r$ right singular vectors of $\Theta_\star$ span, playing $u$ orthogonal to this subspace gives us no additional information about $\Theta_\star$; in this case we should instead play $u$ aligned with this subspace. This is precisely what the first term in (4.3) quantifies, while the second term reflects the fact that we must also estimate the subspace spanned by the top-$r$ right singular vectors of $\Theta_\star$, for which playing inputs isotropically is optimal.

In general, as we do not know $\Theta_\star$, we do not know $V_0$, and so cannot directly compute inputs minimizing (4.3). To circumvent this, we consider the following iterative procedure, which alternates between obtaining an estimate of $\Theta_\star$, $\widehat{\Theta}$, and then playing the inputs that would minimize the estimation error—minimize (4.3)—if $\widehat{\Theta}$ were the true parameter. We present this procedure in Algorithm 1.

---

**Algorithm 1** Active Estimation of Low-Rank Matrices

1: **input:** horizon $N$, feasible inputs $\mathcal{U}$, rank $r$, feasible set $\mathcal{K}$
2: $\widehat{\Theta}_1 \leftarrow I$, $\mathfrak{D} \leftarrow \emptyset$
3: **for** $\ell = 1, 2, 3, \ldots, \lceil \log_2 N \rceil$ **do**
4:      Let $\widehat{V}_0$ denote the top-$r$ right singular vectors of $\widehat{\Theta}_\ell$ and $\Lambda(\lambda) := \sum_{u \in \mathcal{U}} \lambda_u u u^\top$, solve:

$$\lambda_\ell^V \leftarrow \arg\min_{\lambda \in \triangle_\mathcal{U}} \mathrm{tr}\big((\widehat{V}_0^\top \Lambda(\lambda) \widehat{V}_0)^\dagger\big), \quad \lambda_\ell^{\mathrm{unif}} \leftarrow \arg\min_{\lambda \in \triangle_\mathcal{U}} \mathrm{tr}\big(\Lambda(\lambda)^\dagger\big)$$

5:      For $2^\ell$ steps, play input $u_n \sim \frac{1}{2}\lambda_\ell^V + \frac{1}{2}\lambda_\ell^{\mathrm{unif}}$, add observations to $\mathfrak{D}$
6:      Update estimate of $\Theta_\star$: $\widehat{\Theta}_{\ell+1} \leftarrow \arg\min_{\Theta \in \mathcal{K}} \sum_{(u,z) \in \mathfrak{D}} \|z - \Theta u\|_{\mathrm{F}}^2$
7: **return** $\widehat{\Theta}_{\ell+1}$.

---

At every iteration $\ell$, Algorithm 1 computes two distributions over inputs: $\lambda_\ell^V$, which targets the top-$r$ right singular vectors of our current estimate of $\Theta_\star$, and $\lambda_\ell^{\mathrm{unif}}$, which plays inputs isotropically, covering all directions. Rather than playing these distributions according to the precise weighting given in (4.3), we instead found it most effective to mix them at an equal rate. As we do not initially know which directions are spanned by the top-$r$ right singular vectors of $\Theta_\star$, $\lambda_\ell^V$ is not guaranteed to target the correct directions, especially in early iterations. $\lambda_\ell^{\mathrm{unif}}$ plays inputs in every direction,

however, and thus, even if $\lambda_\ell^V$ is not aligned to the top-$r$ right singular vectors of $\Theta_\star$, will ensure sufficient energy is still being played in the correct directions to allow for learning. Given this, we increase the weight of playing $\lambda_\ell^{\mathrm{unif}}$ relative to that prescribed by (4.3).

Note that the computation of the optimal inputs is a form of *A-optimal experiment design* [79], which in general can be efficiently solved by, for example, the Frank-Wolfe algorithm [80]. Furthermore, efficient procedures for solving nuclear-norm regression problems exist, allowing us to estimate $\widehat{\Theta}_{\ell+1}$ on line 6 efficiently [81]. We remark that Algorithm 1 takes as input $r$, the rank of $\Theta_\star$, and $\mathcal{K}$, which requires knowledge of $\|\Theta_\star\|_*$. In general, when these quantities are unknown, they can be chosen via standard cross-validation procedures.

We emphasize again that the setting considered here corresponds precisely to the setting considered in Section 3.2 with $\Theta_\star \leftarrow H$, $u_n$ the input stimulation patterns, and $z_n$ the observed neural response to input $u_n$. As such, if the causal connectivity $H$ is low rank, Algorithm 1 and the preceding results provide a methodology to select input stimuli to most efficiently estimate $H$. In the following section, we will apply this to our photostimulation datasets.

# 5 Active Learning for Estimating Neural Population Dynamics

We return now to the problem of photostimulus design for learning neural population dynamics, and seek to apply the insights of Section 4 to this setting. We present two sets of experiments. In Section 5.1 we use real data to fit a model of the population dynamics, treat this fitted model as a *simulator* for the true dynamics, and then demonstrate that we can learn the causal connectivity matrix $H$ of this simulator faster using active inputs versus passive inputs. Then, in Section 5.2 we split our real data into 750ms long trials of (stimulus, response) pairs (see Section 3) and demonstrate that our active learning algorithm is able to improve the performance of learning dynamical models on real data by adaptively selecting which trials to observe, training a model on the observed trials, and evaluating on a hold-out set of unseen trials. Here we find that our approach is able to learn an accurate model of the dynamics more quickly than non-adaptive approaches.

## 5.1 Active Learning on Data-Driven Neural Population Dynamics Simulator

In Section 3.1, we demonstrated that photostimulation data can be effectively reconstructed using an AR-$k$ dynamics model. Given the effectiveness of these models at fitting our data, in this section we treat them as a simulated representation of our true dynamics, allowing us to query them arbitrarily as a stand-in for the ground truth dynamics, and seek to determine whether carefully choosing the photostimulation pattern allows for efficient estimation of the causal connectivity matrix $H$.

**Experiment Details.** To obtain models of the population dynamics to use for simulation, we fit an AR-$k$ model to each dataset as described in Section 3.1. In all cases we use an AR-$k$ model with order $k = 4$. We do one run of the experiments using low-rank model parameters $UV^\top$ with rank $r = 15$, and then repeat the experiments using $r = 35$. In each case, we simulate $N = 10000$ trials, where each trial corresponds to applying a photostimulus and observing the response for $\tau = 15$ timesteps, simulating our true data generation process. To simulate measurement noise and other trial-to-trial variability in neural responses, we corrupt the observations with Gaussian random noise. Motivated by the empirically observed fast decay of population dynamics in our datasets, we reset the initial state of the simulator at each new trial.

In practice, both the magnitude of the stimuli and number of neurons stimulated at each timestep are constrained by the photostimulation platform. To reflect this limitation in our simulator, we constrain our inputs to lie in $[0, 1]$, and also impose a sparsity penalty. Precisely, we choose the input set $\mathcal{U}$ in Algorithm 1 to be $\mathcal{U} := \{u \in [0,1]^d : \|u\|_1 \le \gamma\}$, for some value $\gamma > 0$ (which we set to $\gamma = 30$). While this does not explicitly constrain inputs to be sparse, it can be efficiently optimized over, and we found in practice that the optimal inputs within this constraint set are in general at least $2\gamma$-sparse. As baseline methods, we consider the following:

- ***Random Stimulation***: At each trial $n$, choose $\gamma$ neurons at random, and set corresponding elements of $u_n$ to 1.

- ***Uniform Stimulation***: Compute $\lambda^{\mathrm{unif}}$ as in Algorithm 1 and play inputs $u_n \sim \lambda^{\mathrm{unif}}$ for all $n$.

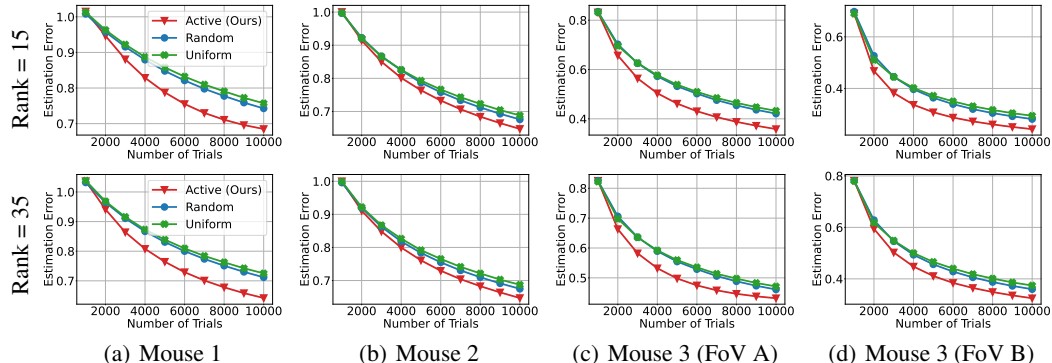

Figure 3: Performance of active stimulation design on estimating learned dynamics model. For each mouse dataset, we fit a low-rank AR-$k$ model as described in Section 3.1 (for ranks of 15 and 35, and $k = 4$). Treating this as a simulator of the true dynamics, we compare our active stimulation design procedure (Active, Algorithm 1) to randomly choosing groups of neurons to excite (Random), and uniformly allocating stimulation across all neurons (Uniform), and plot how effectively each is able to estimate the connectivity of the simulator dynamics. For each figure and method we average over 20 trials, and plot the mean performance with error bars denoting 1 standard error (note that the error bars are barely visible as the standard deviation is very small).

Our goal is to estimate the causal connectivity matrix $H$ induced by our learned dynamics (see Section 3.2). In practice, we are most interested in estimating the *off-diagonal* elements of $H$, as these correspond to causal interactions between different neurons. To this end, we consider the error metric $\frac{\|M \odot (H - \widehat{H})\|_{\mathrm{F}}}{\|M \odot H\|_{\mathrm{F}}}$, for $\widehat{H}$ our estimate of $H$, $M$ a matrix with all entries 1 except its diagonal, which is 0, and $\odot$ element-wise multiplication.

**Experiment Results.** We present our results in Figure 3. As can be seen, across all learned simulators and rank levels, our active learning approach yields a non-trivial gain over both baseline approaches. In particular, on Mouse 1 and both datasets for Mouse 3, we observe a gain of between 1.5-2× over baselines—that is, to achieve a given estimation error, our approach requires between 1.5-2× fewer samples than baseline methods. This demonstrates the effectiveness of our active learning procedure for estimating low-rank matrices—our method is able to exploit the low-rank structure present in the underlying dynamics to speed up estimation, as compared to methods which do not take into account this structure. Furthermore, it shows that on a realistic simulation of neural population dynamics, we can effectively design stimuli to speed up the estimation of the dynamics.

### 5.2 Active Ranking of Real Data Observations

As described in Section 3, each of our datasets consist of roughly 2000 (stimulus, response) trials. In an online photostimulation experiment, we would choose the photostimulus actively for each trial. Here we seek to simulate this process using real experimental data, but offline, by choosing the *ordering* of the trials available in our pre-collected datasets. This serves as a testbed for active learning procedures: if we can more efficiently learn models in this offline setting, that is a strong indication that we should also see gains in online experiments. Indeed, those gains may be even greater online because in our offline setting we are severely restricted to choosing from only 100 candidate stimulation patterns. Thus, we interpret the results in this section as a *lower bound* on the performance we might expect online.

To validate this approach, we randomly choose 20 (out of the 100 total) unique photostimulation patterns and set aside a test set containing all 20 repeated trials of those photostimuli. This creates an 80%/20% train-test split of non-overlapping stimulus patterns. For $\mathfrak{D}_{\mathrm{train}}$ and $\mathfrak{D}_{\mathrm{test}}$ our train and test datasets, respectively, we consider the following query model:

---

1: $\mathfrak{D} \leftarrow \emptyset$
2: **for** trials $n = 1, 2, \ldots, |\mathfrak{D}_{\mathrm{train}}|$ **do**
3:     Choose input trajectory $\tau \in \mathfrak{D}_{\mathrm{train}}$, set $\mathfrak{D} \leftarrow \mathfrak{D} \cup \{\tau\}$, $\mathfrak{D}_{\mathrm{train}} \leftarrow \mathfrak{D}_{\mathrm{train}} \backslash \{\tau\}$
4:     Estimate model using data in $\mathfrak{D}$, and compute prediction MSE of model on $\mathfrak{D}_{\mathrm{test}}$

---

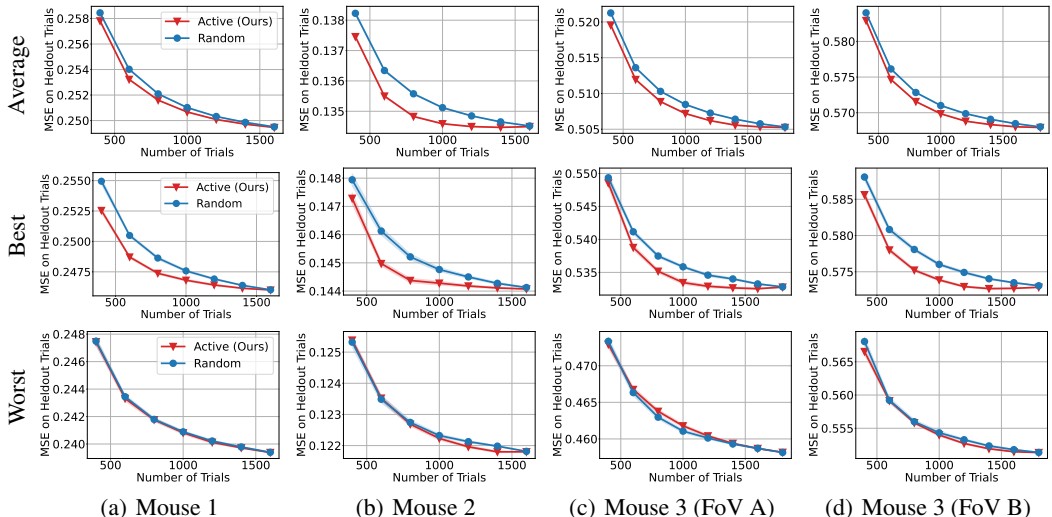

Figure 4: Performance of active learning estimating photostimulation response on held-out trials. Each mouse dataset is split into trials corresponding to a stimulus-response pair, and we consider how these trials might be ordered to obtain more effective estimates with fewer training data trials, simulating the active learning process. Our approach (Active) is motivated by the low-rank excitation criteria of Algorithm 1 (see Appendix B.4 for more details) and we compare with randomly choosing which trial to observe next (Random). We plot the accuracy of the learned model in predicting neural responses on held-out test trials. We consider 20 different train-test splits (with 20 trials per split), and include plots of average performance across these splits, as well as splits where Active has the largest and smallest improvement over Random. We plot error bars denoting 1 standard error (note again that the error bars are barely visible as the standard deviation is very small).

We fit a dynamics model to the current set of observed trials, as described in Section 3.1, and use this model to predict the response of the true system on the held-out test inputs, computing the mean-squared error of these predictions as our metric. We apply a variant of Algorithm 1, described in more detail in Appendix B.4, and adapted to the query model above. In particular, to apply Algorithm 1 to learning a full dynamical system, we choose our inputs to target the right singular vectors of $B_s$ in (3.1). As a baseline method, we consider the procedure which randomly chooses an unobserved segment from $\mathfrak{D}_{\text{train}}$ at each iteration.

We run the above experiment for 20 different randomly generated train-test splits on each dataset, and present our results in Figure 4, providing the results for the average performance over the train-test splits, as well as the best- and worst-case splits for active learning performance. As these results illustrate, though active learning does not give a substantial gain in all cases, in many cases it is able to give a gain of up to a factor of $2\times$ in the number of samples required over the random baseline, and in the worst case, matches the baseline performance. This further confirms that taking into account low-rank structure when choosing which measurements to take can improve estimation rates, and, we believe, is a strong indicator that our active learning procedure would speed up estimation of neural population dynamics in online settings.

## 6 Discussion

In this work, we have developed a principled approach to active learning of photostimulation inputs for the identification of neural population dynamics and connectivity. We discuss three limitations of our approach, which each suggest potential future directions. First, we have considered active learning of the causal connectivity matrix and minimization of prediction error, both uniformly across all recorded neurons. Future work may focus on more specific scenarios, such as targeting particular dimensions of the neural activity space or changes in connectivity due to learning. Second, while we found that linear dynamics fit our data remarkably well, this may not always be the case. Does our methodology effectively scale to nonlinear dynamics? Finally, our real-data experiments were performed offline. Future work may explore running our algorithm online during closed-loop photostimulation experiments.

## Acknowledgments

This work was supported by NSF DMR award 2308979 to the University of Washington Materials Science Research Center (AW & KJ), the Shanahan Foundation Fellowship (LM & MSB), the Paul G. Allen Foundation (MR, KS, KD & MDG), NIH award R00-MH121533 (MDG), NSF CCF award 2007036 (KJ), and NSF CAREER award 2141511 (KJ).

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

# A Proof of Theorem 1

As $\widehat{\Theta}$ is a minimizer, we have that $\|\Phi(\widehat{\Theta}) - z\|_2^2 \leq \|\Phi(\Theta_\star) - z\|_2^2$. Consequently,

$$\|\Phi(\Theta_\star) - z\|_2^2 < \min_{\Theta_\star + \Delta \in \mathcal{K} : \|\Delta\|_F \geq \rho} \|\Phi(\Theta_\star + \Delta) - z\|_2^2 \implies \widehat{\Theta} \notin \{\Theta_\star + \Delta \in \mathcal{K} : \|\Delta\|_F \geq \rho\}$$

$$\implies \|\widehat{\Theta} - \Theta_\star\|_F < \rho.$$

Thus, our strategy will attempt to find a minimum $\rho > 0$ that makes the first expression true. Equivalently, we show that the following quantity is non-positive

$$\|\Phi(\Theta_\star) - z\|_2^2 - \min_{\Theta_\star + \Delta \in \mathcal{K} : \|\Delta\|_F \geq \rho} \|\Phi(\Theta_\star + \Delta) - z\|_2^2$$

$$= -\Big( \min_{\Theta_\star + \Delta \in \mathcal{K} : \|\Delta\|_F \geq \rho} 2\langle \Phi(\Delta), \Phi(\Theta_\star) - z \rangle + \|\Phi(\Delta)\|_2^2 \Big)$$

$$= \max_{M + \Delta \in \mathcal{K} : \|\Delta\|_F \geq \rho} 2\langle \Phi(\Delta), \eta \rangle - \|\Phi(\Delta)\|_2^2$$

$$= \max_{t \geq \rho} \max_{M + \Delta \in \mathcal{K} : \|\Delta\|_F = t} 2\langle \Phi(\Delta), \eta \rangle - \|\Phi(\Delta)\|_2^2.$$

Intuitively, when $t$ is large, the quadratic term will dominate the inner product making the entire expression non-positive. We will employ a geometric fact about the nuclear norm ball.

**Lemma A.1.** *Let* $\mathcal{K} = \{\Theta : \|\Theta\|_* \leq \|\Theta_\star\|_*\}$. *If* $\Theta_\star + \Delta \in \mathcal{K}$ *then* $\|P_\perp(\Delta)\|_* \leq -\langle \Delta, UV^\top \rangle \leq \|P_\parallel(\Delta)\|_F$.

*Proof.* If $\|\Theta_\star + \Delta\|_* > \|\Theta_\star\|_*$ then $\Theta_\star + \Delta \notin \mathcal{K}$. By the convexity of the nuclear norm ball, we have that

$$\|\Theta_\star + \Delta\|_* \geq \|\Theta_\star\|_* + \langle \Delta, UV^\top \rangle + \langle \Delta, W \rangle \qquad \forall W : W = P_\perp(W), \|W\|_2 \leq 1.$$

Consequently, as the dual norm to $\|\cdot\|_2$ is the nuclear norm $\|\cdot\|_*$ we have

$$\|\Theta_\star + \Delta\|_* \geq \|\Theta_\star\|_* + \langle \Delta, UV^\top \rangle + \|P_\perp(\Delta)\|_*.$$

Thus, if $\langle \Delta, UV^\top \rangle + \|P_\perp(\Delta)\|_* > 0$ then $\Theta_\star + \Delta \notin \mathcal{K}$. Consequently, if $\Theta_\star + \Delta \in \mathcal{K}$ then $\langle \Delta, UV^\top \rangle + \|P_\perp(\Delta)\|_* \leq 0$. $\qquad\square$

Recalling that $\|M\|_F \leq \|M\|_*$ for any matrix $M$, an interesting consequence of the above lemma is that $\|P_\perp(\Delta)\|_F \leq \|P_\parallel(\Delta)\|_F$ which implies $\|P_\parallel(\Delta)\|_F^2 \leq \|\Delta\|_F^2 \leq 2\|P_\parallel(\Delta)\|_F^2$. That is, the total error is dominated by the error in the tanget space of $\Theta_\star$.

Applying this lemma, we have that

$$\max_{M + \Delta \in K : \|\Delta\|_F = t} 2\langle \Phi(\Delta), \eta \rangle - \|\Phi(\Delta)\|_2^2 \leq \max_{\Delta : \|P_\perp(\Delta)\|_* \leq \|P_\parallel(\Delta)\|_F, \|\Delta\|_F = t} 2\langle \Phi(\Delta), \eta \rangle - \|\Phi(\Delta)\|_2^2$$

$$= \max_{\Delta : \|P_\perp(\Delta)\|_* \leq \|P_\parallel(\Delta)\|_F, \|\Delta\|_F = t} 2\langle \Phi(P_\parallel(\Delta)) + \Phi(P_\perp(\Delta)), \eta \rangle - \|\Phi(\Delta)\|_2^2$$

$$\leq \max_{\Delta : \|P_\perp(\Delta)\|_* \leq \|P_\parallel(\Delta)\|_F, \|\Delta\|_F = t} 2\langle \Phi(P_\parallel(\Delta)), \eta \rangle - \|\Phi(\Delta)\|_2^2$$

$$+ \max_{\Delta : \|P_\perp(\Delta)\|_* \leq \|P_\parallel(\Delta)\|_F, \|\Delta\|_F = t} 2\langle \Phi(P_\perp(\Delta)), \eta \rangle$$

where the equality uses the fact that $\Delta = P_\parallel(\Delta) + P_\perp(\Delta)$ and the linearity of $\Phi$. For any $v \in \mathbb{R}^N$ we have

$$\langle \Phi(\Delta), v \rangle = \langle \Delta, \sum_{n=1}^N X_n v_n \rangle =: \langle \Delta, \Phi^*(v) \rangle$$

where $\Phi^*$ denotes the adjoint of $\Phi$. We also recognize that the operator $\Phi^*\Phi : \mathbb{R}^{d_1 \times d_2} \to \mathbb{R}^{d_1 \times d_2}$ is also linear, defined as $\Phi^*\Phi(M) = \Phi^*(\{\langle X_n, M \rangle\}_n) = \sum_{n=1}^N X_n \langle X_n, M \rangle$. Consequently $(\Phi^*\Phi)^{1/2}$ is well-defined and is the same operator as $\Phi^*\Phi$ after taking the square root of its eigenvalues.

The next three lemmas bound the two terms of above. Combining them yields the result of the theorem.

**Lemma A.2.** *With probability at least $1 - \delta$ we have*

$$\max_{\Delta: \|P_\perp(\Delta)\|_* \le \|P_\parallel(\Delta)\|_F, \|\Delta\|_F = t} 2\langle \Phi(P_\perp(\Delta)), \eta \rangle \le 2t \|P_\perp(\Phi^*\Phi)^{1/2}\|_{\mathrm{op}}(\sqrt{d_1} + \sqrt{d_2} + \sqrt{2\log(1/\delta)}).$$

*Proof.* Computing this term amounts to bounding a Gaussian width. Begin by recognizing that by the non-expansive property of projections, $\|P_\parallel(\Delta)\|_F \le \|\Delta\|_F \le t$ which results in the simplification:

$$\max_{\Delta: \|P_\perp(\Delta)\|_* \le \|P_\parallel(\Delta)\|_F, \|\Delta\|_F = t} 2\langle \Phi(P_\perp(\Delta)), \eta \rangle \le \max_{\Delta: \|P_\perp(\Delta)\|_* \le t} 2\langle \Phi(P_\perp(\Delta)), \eta \rangle$$

$$= \max_{\Delta: \|P_\perp(\Delta)\|_* \le t} 2\langle \sum_{n=1}^{N} \eta_n X_n, P_\perp(\Delta) \rangle$$

$$= \max_{\Delta: \|P_\perp(\Delta)\|_* \le t} 2\langle P_\perp(\sum_{n=1}^{N} \eta_n X_n), P_\perp(\Delta) \rangle$$

$$\le 2t \|P_\perp(\sum_{n=1}^{N} \eta_n X_n)\|_{\mathrm{op}}$$

using the fact that the operator norm and nuclear norm are dual to each other, and that the projection $P_\perp$ is non-expansive. Note that

$$\sum_{n=1}^{N} \eta_n X_n = \mathrm{mat}(\sum_{n=1}^{N} \eta_n \mathrm{vec}(X_i)) = \mathrm{mat}((\sum_{n=1}^{N} \mathrm{vec}(X_n)\mathrm{vec}(X_n)^\top)^{1/2}\mathrm{vec}(\eta')) =: (\Phi^*\Phi)^{1/2}(\eta)$$

where $\eta' \in \mathbb{R}^{d_1 \times d_2}$ with $\eta'_{i,j} \sim \mathcal{N}(0,1)$. We then observe that

$$\|P_\perp(\sum_{n=1}^{N} \eta_n X_n)\|_{\mathrm{op}} = \|P_\perp(\mathrm{mat}((\sum_{n=1}^{N} \mathrm{vec}(X_n)\mathrm{vec}(X_n)^\top)^{1/2}\mathrm{vec}(\eta')))\|_{\mathrm{op}}$$

$$\le \|P_\perp(\Phi^*\Phi)^{1/2}\|_{\mathrm{op}}\|\eta'\|_{\mathrm{op}}$$

which completes the proof of the first claim. Recognizing that $\|\eta'\|_{\mathrm{op}}$ is just the largest singular value of a Gaussian matrix, we find that $\mathbb{E}[\sup_{\|u\|_2 \le 1, \|v\|_2 \le 1}\langle uv^\top, \eta \rangle] \le \sqrt{d_1} + \sqrt{d_2}$ by Exercise 5.14 of [52]. Applying a sub-Gaussian tail bound completes the proof. $\square$

**Lemma A.3.** *Define $\mu := \|(\Phi^*\Phi)^{1/2}((P_\parallel\Phi^*\Phi P_\parallel)^\dagger)^{1/2}\|_{\mathrm{op}}$. Then*

$$\max_{\Delta: \|P_\perp(\Delta)\|_* \le \|P_\parallel(\Delta)\|_F, \|\Delta\|_F = t} 2\langle \Phi(P_\parallel(\Delta)), \eta \rangle - \|\Phi(\Delta)\|_2^2$$

$$\le \max_{\Delta: \|P_\parallel(\Delta)\|_F \ge t/\sqrt{2}} 2\langle \Phi(P_\parallel(\Delta)), \eta \rangle - (1-\mu)\|\Phi(P_\parallel(\Delta))\|_2^2$$

*Proof.* Recognizing that $\Phi(\Delta) \in \mathbb{R}^N$ we have

$$\|\Phi(\Delta)\|_2^2 = \|\Phi(P_\parallel(\Delta) + P_\perp(\Delta))\|_2^2$$

$$= \|\Phi(P_\parallel(\Delta)) + \Phi(P_\perp(\Delta))\|_2^2$$

$$= \|\Phi(P_\parallel(\Delta))\|_2^2 + \|\Phi(P_\perp(\Delta))\|_2^2 + 2\langle \Phi(P_\parallel(\Delta)), \Phi(P_\perp(\Delta)) \rangle.$$

To aid in readability, we make a number of notational modifications. First, we drop parentheses so that $\Phi(P_\parallel(\Delta))$ is just notated as $\Phi P_\parallel \Delta$. Second, we define $M^{\dagger/2} := (M^\dagger)^{1/2}$ where $M^\dagger$ is the pseudoinverse. If $\Phi^*\Phi$ is invertible restricted to the range of $P_\parallel$, then $P_\parallel\Delta = (P_\parallel\Phi^*\Phi P_\parallel)^{\dagger/2}(\Phi^*\Phi)^{1/2}P_\parallel\Delta$ for all $\Delta$. Thus,

$$|\langle \Phi(P_\parallel(\Delta)), \Phi(P_\perp(\Delta)) \rangle| = |\langle \Phi P_\parallel\Delta, \Phi P_\perp\Delta \rangle|$$

$$= |\langle P_\parallel\Delta, (\Phi^*\Phi)P_\perp\Delta \rangle|$$

$$= |\langle (\Phi^*\Phi)^{1/2}P_\parallel\Delta, (\Phi^*\Phi)^{1/2}P_\perp\Delta \rangle|$$

$$= |\langle (\Phi^*\Phi)^{1/2}P_\parallel(P_\parallel\Phi^*\Phi P_\parallel)^{\dagger/2}(\Phi^*\Phi)^{1/2}P_\parallel\Delta, (\Phi^*\Phi)^{1/2}P_\perp\Delta \rangle|$$

$$\le \|(\Phi^*\Phi)^{1/2}(P_\parallel\Phi^*\Phi P_\parallel)^{\dagger/2}(\Phi^*\Phi)^{1/2}P_\parallel\Delta\|_F \, \|(\Phi^*\Phi)^{1/2}P_\perp\Delta\|_F$$

$$\le \|(\Phi^*\Phi)^{1/2}(P_\parallel\Phi^*\Phi P_\parallel)^{\dagger/2}\|_{\mathrm{op}} \, \|(\Phi^*\Phi)^{1/2}P_\parallel\Delta\|_F \, \|(\Phi^*\Phi)^{1/2}P_\perp\Delta\|_F$$

where the last two lines follow from Cauchy-Schwartz. Observe that for $\mu < 1$ we have

$$a^2 + b^2 - 2ab\mu = (1-\mu)a^2 + (1-\mu)b^2 + \mu a^2 + \mu b^2 - 2ab\mu$$
$$= (1-\mu)a^2 + (1-\mu)b^2 + \mu(a-b)^2$$
$$\geq (1-\mu)a^2.$$

Thus, if $\mu := \|(\Phi^*\Phi)^{1/2}(P_\|\Phi^*\Phi P_\|)^{\dagger/2}\|_{\mathrm{op}}$ then

$$\|\Phi(\Delta)\|_2^2 \geq \|\Phi(P_\|(\Delta))\|_2^2 + \|\Phi(P_\perp(\Delta))\|_2^2 - 2|\langle \Phi(P_\|(\Delta)), \Phi(P_\perp(\Delta))\rangle|$$
$$\geq \|\Phi(P_\|(\Delta))\|_2^2 + \|\Phi(P_\perp(\Delta))\|_2^2 - 2\mu\|(\Phi^*\Phi)^{1/2}P_\|\Delta\|_{\mathrm{F}}\,\|(\Phi^*\Phi)^{1/2}P_\perp\Delta\|_{\mathrm{F}}$$
$$\geq (1-\mu)\|\Phi(P_\|(\Delta))\|_2^2$$

Thus,

$$\max_{\Delta:\|P_\perp(\Delta)\|_*\leq\|P_\|(\Delta)\|_{\mathrm{F}}, \|\Delta\|_{\mathrm{F}}=t} 2\langle\Phi(P_\|(\Delta)),\eta\rangle - \|\Phi(\Delta)\|_2^2$$

$$\leq \max_{\Delta:\|P_\perp(\Delta)\|_*\leq\|P_\|(\Delta)\|_{\mathrm{F}}, \|\Delta\|_{\mathrm{F}}=t} 2\langle\Phi(P_\|(\Delta)),\eta\rangle - (1-\mu)\|\Phi(P_\|(\Delta))\|_2^2$$

$$\leq \max_{\Delta:\|P_\perp(\Delta)\|_{\mathrm{F}}\leq\|P_\|(\Delta)\|_{\mathrm{F}}, \|\Delta\|_{\mathrm{F}}=t} 2\langle\Phi(P_\|(\Delta)),\eta\rangle - (1-\mu)\|\Phi(P_\|(\Delta))\|_2^2$$

$$\leq \max_{\Delta:\|P_\|(\Delta)\|_{\mathrm{F}}\geq t/\sqrt{2}} 2\langle\Phi(P_\|(\Delta)),\eta\rangle - (1-\mu)\|\Phi(P_\|(\Delta))\|_2^2$$

where we've used the facts that $\|\cdot\|_* \leq \|\cdot\|_{\mathrm{F}}$ and $t^2 = \|\Delta\|_{\mathrm{F}}^2 = \|P_\|(\Delta)\|_{\mathrm{F}}^2 + \|P_\perp(\Delta)\|_{\mathrm{F}}^2 \leq 2\|P_\|(\Delta)\|_{\mathrm{F}}^2$. $\qquad\square$

**Lemma A.4.** *Let $K = \min\{\log_2(N), d_1 d_2\}$. Then for any $\alpha > 0$, if*

$$t \geq \frac{1}{1-\mu}\sqrt{16\mathrm{tr}\left((P_\|\Phi^*\Phi P_\|)^\dagger\right) + 32\|(P_\|\Phi^*\Phi P_\|)^\dagger\|_{\mathrm{op}}\log(K/\delta)} + \frac{2\alpha\|(P_\|\Phi^*\Phi P_\|)^\dagger\|_{\mathrm{op}}}{1-\mu}$$

*then with probability at least $1 - \delta$ we have*

$$\alpha t + \max_{\Delta:\|P_\|(\Delta)\|_{\mathrm{F}}\geq t/\sqrt{2}} 2\langle\Phi(P_\|(\Delta)),\eta\rangle - (1-\mu)\|\Phi(P_\|(\Delta))\|_2^2 \leq 0.$$

*Proof.* The linear operator $\Phi P_\| : \mathbb{R}^{d_1\times d_2} \to \mathbb{R}^N$ can be decomposed as $\Phi P_\| = \sum_{n=1}^N \beta_n w_n \psi_n$ where $\{w_n\}_n$ are orthonormal on $\mathbb{R}^T$, $\{\psi_n\}_n$ are orthonormal linear operators on $\mathbb{R}^{d_1\times d_2}$, and $\beta_n \geq 0$ are decreasing. For $k = 0, 1, \ldots, \min\{\log_2(N), d_1 d_2\} - 1$ let $W_k = [w_{2^k}, \ldots, w_{2^{k+1}-1}]$ so that

$$\beta_{2^k} = \max_{\|\Delta\|_{\mathrm{F}}=1,\|u\|_2=1} u^\top W_k^\top \Phi P_\|(\Delta) \geq \min_{\|\Delta\|_{\mathrm{F}}=1,\|u\|_2=1} u^\top W_k^\top \Phi P_\|(\Delta) \geq \beta_{2^{k+1}}$$

Then

$$\max_{\Delta:\|P_\|(\Delta)\|_{\mathrm{F}}\geq t/\sqrt{2}} 2\langle\Phi(P_\|(\Delta)),\eta\rangle - (1-\mu)\|\Phi(P_\|(\Delta))\|_2^2$$

$$= \max_{\Delta:\|P_\|(\Delta)\|_{\mathrm{F}}\geq t/\sqrt{2}} \sum_{k=0}^K 2\langle W_k W_k^\top \Phi(P_\|(\Delta)),\eta\rangle - (1-\mu)\|W_k W_k^\top \Phi(P_\|(\Delta))\|_2^2$$

$$= \max_{\Delta:\|P_\|(\Delta)\|_{\mathrm{F}}\geq t/\sqrt{2}} \sum_{k=0}^K 2\langle W_k^\top \Phi(P_\|(\Delta)), W_k^\top\eta\rangle - (1-\mu)\|W_k^\top \Phi(P_\|(\Delta))\|_2^2$$

$$\leq \sum_{k=0}^K \max_{\Delta:\|P_\|(\Delta)\|_{\mathrm{F}}\geq t/\sqrt{2}} 2\|W_k^\top \Phi(P_\|(\Delta))\|_2\,\|W_k^\top\eta\|_2 - (1-\mu)\|W_k^\top \Phi(P_\|(\Delta))\|_2^2$$

$$\leq \sum_{k=0}^K t\sqrt{2}\beta_{2^{k+1}}\|W_k^\top\eta\|_2 - \frac{t^2(1-\mu)}{2}\beta_{2^{k+1}}^2$$

where the first inequality holds by Cauchy-Schwartz, and the second holds for all $t \geq \max_{k=0,\ldots,K} \frac{\|W_k^\top \eta\|_2 2\sqrt{2}}{(1-\mu)\beta_{2^{k+1}}}$. Moreover, $t\sqrt{2}\beta_{2^{k+1}}\|W_k^\top \eta\|_2 - \frac{t^2(1-\mu)}{2}\beta_{2^{k+1}}^2 \leq -\alpha t$ if

$$t \geq \max_{k=0,\ldots,K} \frac{\|W_k^\top \eta\|_2 2\sqrt{2}}{(1-\mu)\beta_{2^{k+1}}} + \max_{k=0,\ldots,K} \frac{2\alpha}{(1-\mu)\beta_{2^{k+1}}^2} = \sqrt{\max_{k=0,\ldots,K} \frac{8\|W_k^\top \eta\|_2^2}{(1-\mu)^2\beta_{2^{k+1}}^2}} + \max_{k=0,\ldots,K} \frac{2\alpha}{(1-\mu)\beta_{2^{k+1}}^2}$$

Note that for $K = \min\{\log_2(N), d_1 d_2\}$ we have

$$\mathbf{P}(\cup_{k=1}^K \{\|W_k^\top \eta\|_2 \geq \sqrt{2^k} + \sqrt{2\log(K/\delta)}\}) \leq \mathbf{P}(\cup_{k=1}^K \{\|W_k^\top \eta\|_2 \geq \mathbb{E}[\|W_k^\top \eta\|_2] + \sqrt{2\log(K/\delta)}\}) \leq \delta.$$

On this good event, we have that

$$\begin{aligned}
\max_{k=0,\ldots,K} \frac{\|W_k^\top \eta\|_2^2}{\beta_{2^{k+1}}^2} &\leq \max_{k=0,\ldots,K} \frac{(\sqrt{2^k} + \sqrt{2\log(K/\delta)})^2}{\beta_{2^{k+1}}^2} \\
&\leq \max_{k=0,\ldots,K} \frac{2^{k+1}}{\beta_{2^{k+1}}^2} + \frac{4\log(K/\delta)}{\beta_{2^{k+1}}^2} \\
&\leq 2\sum_{n=1}^N \frac{1}{\beta_n^2} + \max_{n=1,\ldots,N} \frac{4\log(K/\delta)}{\beta_n^2} \\
&= 2\mathrm{tr}\left((P_\| \Phi^* \Phi P_\|)^\dagger\right) + 4\|(P_\| \Phi^* \Phi P_\|)^\dagger\|_{\mathrm{op}} \log(K/\delta)
\end{aligned}$$

where we use the fact that

$$\sum_{n=1}^N \frac{1}{\beta_n^2} = \sum_{k=0}^K \sum_{n=2^k}^{2^{k+1}-1} \frac{1}{\beta_n^2} \geq \sum_{k=0}^{K-1} \frac{2^k}{\beta_{2^{k+1}}^2} \geq \max_{k=0,\ldots,K-1} \frac{2^k}{\beta_{2^{k+1}}^2}.$$

$\square$

# B    Additional Details on Experiments

## B.1    Further Details on Dataset

The photostimulation data were collected from transgenic reporter mice Ai229, which express Cre-recombinase-dependent cytosolic GCaMP6m and soma-targeted ChRmine, crossed with the Vglut1-cre mouse line. Imaging and photostimulation experiments were performed on a Bergamo (Thorlabs) microscope equipped with a 16x (0.8 NA) Nikon objective. Post-hoc motion correction and neuron segmentation were performed with the Suite2p package [82] (https://github.com/MouseLand/suite2p).

## B.2    Further Details on Experiment of Section 3.1

We split each of our photostimulation datasets into non-overlapping training and test datasets. All models were trained exclusively using the training dataset and were then evaluated (as shown in Figure 2) using the test dataset. To build our test datasets, we randomly chose 5 (out of the 100 total) unique photostimulation patterns and then included all 70-timestep windows about each of the 20 instances of those 5 unique photostimuli. The resulting test set amounted to $\sim$20% of each dataset. In Figure 2, all models were evaluated using these 70-timestep test sequences of the form $\{y_t, u_t\}_{t=1}^{70}$, where $y_t \in R^d$ is the recorded neural activity and $u_t \in R^d$ is the photostimulation delivered at time $t$. During evaluation on a given test window, all models were provided $\{y_t\}_{t=1}^4$ and $\{u_t\}_{t=1}^{70}$ to predict $\{y_t\}_{t=5}^{70}$.

**Autoregressive-k models:** We fit the full-rank AR-$k$ models to training datasets via linear regression by expressing

$$y_{t+1} = \sum_{s=0}^{k-1} A_s y_{t-s} + \sum_{s=0}^{k-1} B_s u_{t-s} + v \tag{B.1}$$

as $Y = XW$, where

$$Y = \begin{bmatrix} y_1 \\ y_2 \\ \vdots \\ y_{T+1} \end{bmatrix} \quad X = \begin{bmatrix} y_0 & y_{-1} & \cdots & y_{1-k} & u_0 & u_{-1} & \cdots & u_{1-k} & 1 \\ y_1 & y_0 & \cdots & y_{2-k} & u_1 & u_0 & \cdots & u_{2-k} & 1 \\ \vdots & \vdots & & \vdots & \vdots & \vdots & & \vdots & 1 \\ y_T & y_{T-1} & \cdots & y_{T+1-k} & u_T & u_{T-1} & \cdots & u_{T+1-k} & 1 \end{bmatrix}$$

$$W = \begin{bmatrix} A_0 \\ A_1 \\ \vdots \\ A_{k-1} \\ B_0 \\ B_1 \\ \vdots \\ B_{k-1} \\ v \end{bmatrix}$$

and the closed-form solution is $\widehat{W} = (X^T X)^{-1} X^T Y$. For the low-rank AR-$k$ models, we fit all parameters via gradient descent using Adam [83] over 100 training epochs with a learning rate of 0.01. Gradient descent was implemented in PyTorch and ran on a single NVIDIA Tesla T4 GPU.

During evaluation of the AR-$k$ models, for each test window we first computed $\widehat{y}_{k+1}$ given $\{y_t, u_t\}_{t=1}^{k}$ using (B.1). Then for all subsequent predictions, on the right-hand side of (B.1) we replaced all instances of $y_t$ with $\widehat{y}_t$ for $t > k$. In this manner, each entire roll-out prediction of $\{y_t\}_{t=k+1}^{70}$ used all photostimulation inputs $\{u_t\}_{t=1}^{70}$, but only the first $k$ timesteps of neural activity $\{y_t\}_{t=1}^{k}$. All AR-$k$ models in this paper used $k = 4$.

**Gated recurrent unit (GRU) networks:** GRU networks were loosely based on the sequential variational autoencoders of [22]. Each model consisted of an encoder GRU network that encodes $k = 4$ initial timesteps of recorded neural activity into a bottlenecked initial state for a decoder GRU network. The decoder then unrolls an entire predicted timeseries of recorded neural activity given (as input) all photostimulation that was delivered over that time period. Model fitting proceeded by optimizing the evidence lower bound (ELBO) with respect to all model parameters. Both encoder and decoder GRUs had 512 hidden units. We used Adam optimization with a learning rate of 0.001 over 4000 training epochs of batch size 100. Models were implemented with PyTorch, and optimized on a single NVIDIA Tesla T4 GPU.

**Evaluation metrics:** We evaluated all models using roll-out predictions on held-out test windows. We quantified performance with mean squared error between recorded and predicted neural activity for each neuron. We also performed thresholded response detection, whereby detections were defined as timesteps at which a given neuron's measured calcium fluorescence exceeded a predefined threshold. To calculate a receiver operator characteristic (ROC) curve, we enumerated a range of thresholds, normalized by the standard deviation of each neuron's empirical activity distribution, and performed threshold detection separately on the real and model-predicted neural activity traces. We then compute the overall false-positive rate and true-positive rate at each threshold level to trace out an ROC curve. We calculate area under the ROC curve (AUROC) to quantify the accuracy of each model.

**Longer roll-out evaluations:** To assess AR-$k$ models' ability to predict over longer time horizons, we implemented another train-test strategy, where the first 80% of timesteps in a recording are used for training, and the last 20% of timesteps (6736 steps) are used for testing. During the test phase, we use the same procedure described above, providing only the $k = 4$ initial timesteps of neural activity and then unrolling predictions over the remainder of this long test window. We report these results in Figure 5.

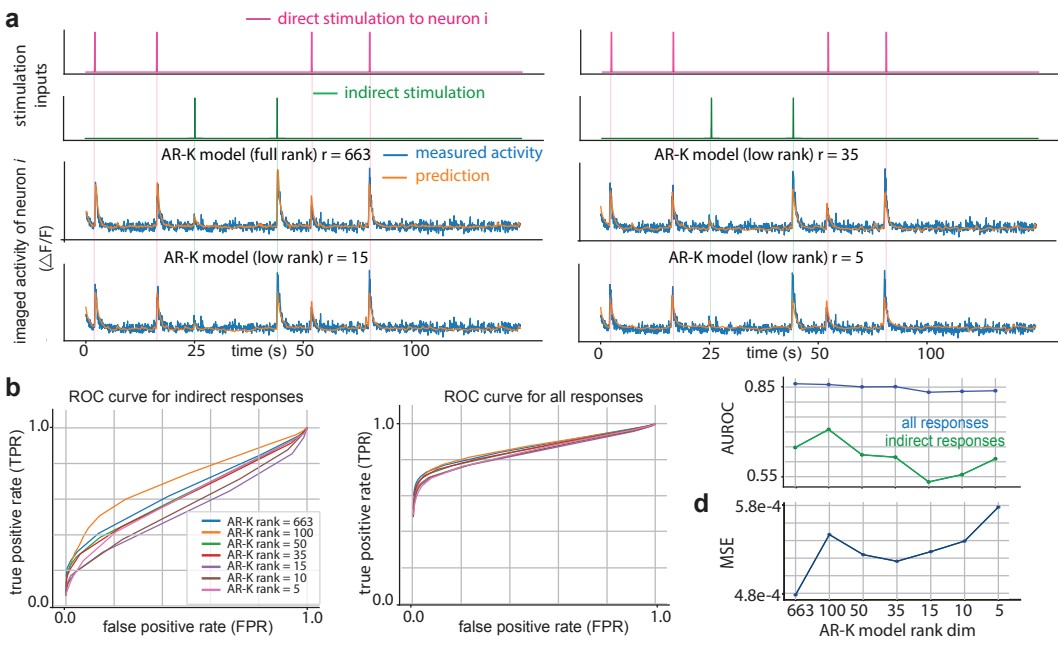

Figure 5: Longer roll-out evaluations. Same format as in Figure 2.

## B.3 Further Details on Experiment of Section 5.1

To fit the $H$ parameter in this experiment, we generate observations as described in Section 5.1. We then estimate $H$ as:

$$\widehat{H} \leftarrow \operatorname*{arg\,min}_{H \in \mathcal{K}} \sum_{(u,z) \in \mathfrak{D}} \|z - Hu\|_{\mathrm{F}}^2$$

for $u$ our input, and $z = \sum_{t=1}^{\tau} x_t$ the observed response, where here $x_t$ are the observations generated from playing input $u$, and $\tau = 15$.

As $\mathcal{K}$ is defined with respect to the nuclear norm of the true parameter, which we do not assume is known, we run each method with a range of possible values for the nuclear-norm constraint, and plot the performance of each method for the constraint value that has minimum error. We state the value of the nuclear-norm constraint used for each plot below:

|  | Active | Random | Uniform |
|---|---|---|---|
| Mouse 1, rank 15 | 10 | 10 | 10 |
| Mouse 1, rank 35 | 10 | 10 | 10 |
| Mouse 2, rank 15 | 5 | 5 | 5 |
| Mouse 2, rank 35 | 5 | 5 | 5 |
| Mouse 3 (FoV A), rank 15 | 25 | 25 | 25 |
| Mouse 3 (FoV A), rank 35 | 25 | 25 | 25 |
| Mouse 3 (FoV B), rank 15 | 50 | 100 | 100 |
| Mouse 3 (FoV B), rank 35 | 100 | 100 | 100 |

Table 1: Nuclear-Norm Constraint Settings for Results of Section 5.1

To choose the input rank of Algorithm 1, we ran our experiment with several different ranks and provide results for the best-performing rank. We found, however, that results are typically robust to the setting of the rank parameter of Algorithm 1, and our choice of $r$ did not significantly impact performance. Furthermore, we believe this could effectively be chosen adaptively. We state our chosen values of $r$ below.

For all experiments, we add observation noise distributed as $\mathcal{N}(0, 0.4 \cdot I)$ to $z = \sum_{t=1}^{\tau} x_t$.

| | Input Rank $r$ |
|---|---|
| Mouse 1, rank 15 | 10 |
| Mouse 1, rank 35 | 10 |
| Mouse 2, rank 15 | 5 |
| Mouse 2, rank 35 | 5 |
| Mouse 3 (FoV A), rank 15 | 25 |
| Mouse 3 (FoV A), rank 35 | 25 |
| Mouse 3 (FoV B), rank 15 | 25 |
| Mouse 3 (FoV B), rank 35 | 50 |

Table 2: Input Rank $r$ for Results of Section 5.1

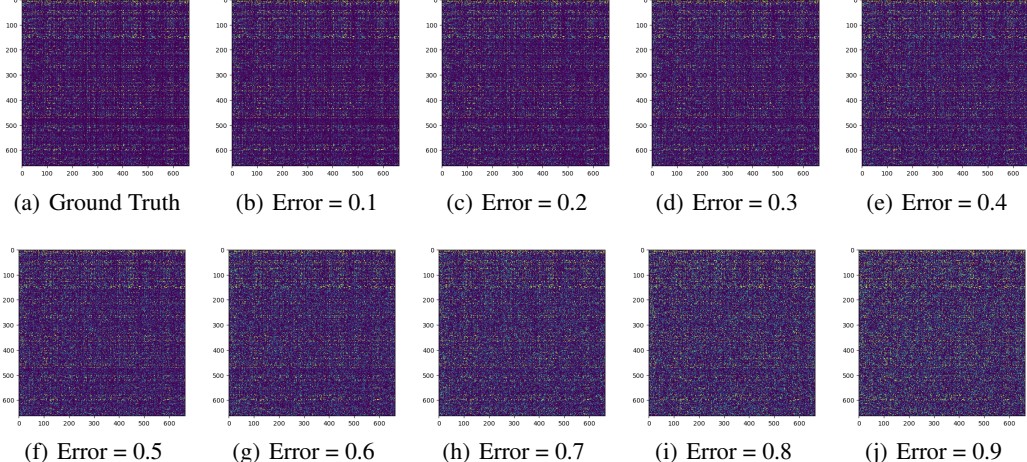

(a) Ground Truth    (b) Error = 0.1    (c) Error = 0.2    (d) Error = 0.3    (e) Error = 0.4

(f) Error = 0.5    (g) Error = 0.6    (h) Error = 0.7    (i) Error = 0.8    (j) Error = 0.9

Figure 6: Causal connectivity matrix for Mouse 3 FoV B with different levels of estimation error (corresponding to Figure 3).

To ground the estimation error values shown in Figure 3, in Figure 6 will illustrate the causal connectivity matrix for Mouse 3 FoV B with different levels of estimation error.

### B.4 Further Details on Experiment of Section 5.2

For this experiment, on the data $\mathfrak{D}$ we observed thus far, we fit the AR-$k$ model described in Section 3.1 with $k = 1$. We found that for this experiment, simply using the least squares estimator with no low-rank penalty produced the best results. We use the same estimation method for both our method and the baseline method.

Given a input response trajectory in the test set, $(x_1, \ldots, x_{15})$, with input $u$, to compute the test MSE, we provide our learned dynamics model with the initial state $x_1$ and input $u$, and then roll this out for 15 timesteps to generate predictions $\widehat{x}_2, \ldots, \widehat{x}_{15}$. Precisely, if $\widehat{A}$ and $\widehat{B}$ are our estimated parameters, we let

$$\widehat{x}_2 = \widehat{A}x_1 + \widehat{B}u,$$
$$\widehat{x}_{t+1} = \widehat{A}\widehat{x}_t, \quad t \geq 1.$$

We the compute the MSE on this segment as:

$$\frac{1}{14} \sum_{t=2}^{15} \|\widehat{x}_t - x_t\|_2^2$$

It is not immediately obvious how to apply Algorithm 1 to this setting, since we must choose each trajectory sequentially, and once we have observed a trajectory it can no longer be chosen again. Rather than solving the optimization of Algorithm 1 to find the best inputs, we instead seek to

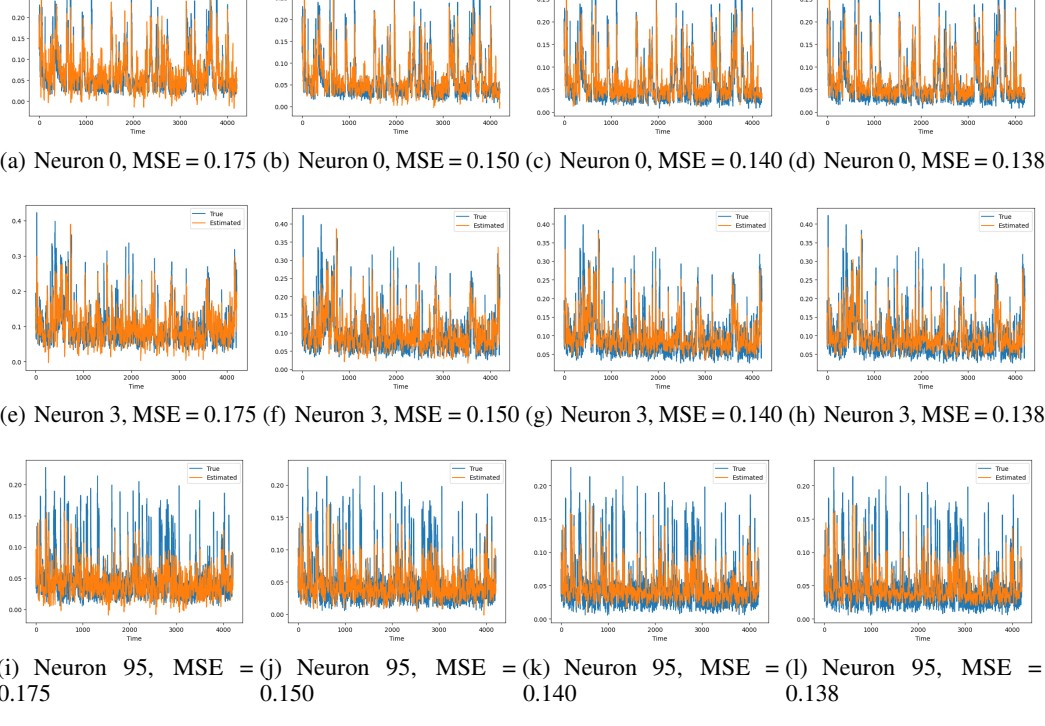

(a) Neuron 0, MSE = 0.175 (b) Neuron 0, MSE = 0.150 (c) Neuron 0, MSE = 0.140 (d) Neuron 0, MSE = 0.138

(e) Neuron 3, MSE = 0.175 (f) Neuron 3, MSE = 0.150 (g) Neuron 3, MSE = 0.140 (h) Neuron 3, MSE = 0.138

(i) Neuron 95, MSE = (j) Neuron 95, MSE = (k) Neuron 95, MSE = (l) Neuron 95, MSE = 0.175 0.150 0.140 0.138

Figure 7: Estimated neural activity vs true neural activity on heldout trials for Mouse 2, Neurons 0, 3, and 95, at different levels of overall MSE on heldout trials (corresponding to Figure 4).

iteratively choose the next input that would maximize "information gain" in some sense. In particular, note that applying the Frank-Wolfe algorithm [80] to the objective, if we have inputs $\mathcal{U}$ available:

$$\min_{\lambda \in \triangle_{\mathcal{U}}} \operatorname{tr}((V^\top \mathbf{\Lambda}(\lambda)V)^{-1}),$$

the update is given by:

$$u_{i+1} = \min_{u \in \mathcal{U}} u^\top V (V^\top \mathbf{\Lambda}(\lambda_i)V)^{-2} V^\top u$$

$$\lambda_{i+1} \leftarrow (1 - \gamma_i)\lambda_i + \gamma_i \mathbb{I}\{u = u_{i+1}\}$$

for learning rate $\gamma_i$.

In this experiment, we simply choose $u_n$ as above, with $\mathcal{U}$ the set of remaining active inputs in $\mathfrak{D}_{\text{train}}$, and $\mathbf{\Lambda}(\lambda_n)$ replaced with $\sum_{s=1}^{n-1} u_s u_s^\top$. This therefore approximates the solution to the experiment design of Algorithm 1, and has the advantage of being very computationally efficient. Furthermore, we set $V$ to be the right singular vectors of $\widehat{B}$. We believe this is reasonable in dynamical system settings with fast decay.

The primary hyperparameter for this experiment is the choice of $r$, the rank of $V$. As in the previous section, we did not find the results particularly sensitive to setting of $r$. For each dataset, we ran with $r \in [25, 50, 75, 100, 125, 150]$, and include results for the best-performing setting.

For both sets of experiments in Section 5, we ran on 56 Intel(R) Xeon(R) CPU E5-2690 v4 @ 2.60GHz CPUs.

To ground the MSE values shown in Figure 4, in Figure 7 we plot the predictions from the estimated model at different MSE values on heldout trials for Mouse 2.

