# OpenReview forum: "Active learning of neural population dynamics using two-photon holographic optogenetics"
_NeurIPS.cc/2024/Conference — NeurIPS 2024 poster_

### Official Review · Reviewer_d6y7 · 2024-06-17

**Soundness:** 4
**Presentation:** 2
**Contribution:** 3
**Rating:** 6
**Confidence:** 3

**Summary:**

The authors develop active learning techniques to design photostimulation experiments combined with 2p imaging to uncover dynamical systems model of brain activity. To this end, the authors employ low rank matrix recovery techniques. They demonstrate their approach on a dataset from mouse motor cortex.

**Strengths:**

* Develops a low-rank matrix estimation framework for dynamical systems of neural activity upon photostimulation.
* Develops an active learning approach to effectively identify components of the system.
* The authors demonstrate the usefulness of their approach using simulated experimental settings, which make use of real data.

**Weaknesses:**

* Fig. 2: It is unclear why the GRU is shown if it is not used later.
* Line 21: I feel like have read the first sentence many times, almost every year at NeurIPS, in papers on computational neuroscience, where dynamics of neural populations is replaced by whatever the paper studies. You could simply start by the third or the forth sentence without any loss in information.
* The exposition in Section 4 is overly technical. It would help if the high level narrative of the paper and the essential steps where explained on a somewhat more intuitive level.
* No errorbars in Figure 3 and 4.

**Questions:**

* Line 138: Why did the authors choose the term Causal connectivity matrix? I don’t understand why H would be a causal estimate. It would be good if the authors could elaborate or choose another term.
* Wouldn’t theorem 2 be sufficient or the theorem from ref 67 for the point the authors want to make? How does Theorem 2 reconcile with the claim of the authors in line 176 that this is the first such bound.
* Line 179: Why is it so important to have an error bound for active learning as compared to other settings?
* Section 4.2: I understand that Algorithm 1 somehow falls out of the theory, but it would be helpful for me if the authors could convey the intuition between \lambda^V and \lambda^{uniform}, and why one wants to mix them.
* Line 205: What is meant by “played isotropically”?
Algorithm 1: What would be a typical range for l (log T)? Is it possible that the 2^l loop in step 5 becomes infeasibly long?
* Line 248: Is this the same T as in Algorithm 1?

**Limitations:**

The limitations are very briefly discussed but sufficient.

---

> ### Author Rebuttal · Authors · 2024-08-07
>
> > Fig. 2: It is unclear why the GRU is shown if it is not used later.
>
> We include results for the GRU model to justify our use of the linear model. In particular, while similar GRU models have often appeared in the computational neuroscience literature (e.g., Pandarinath et al., Nature Methods, 2018), our results show that on this dataset, we can obtain a more effective fit using a linear model, which validates our use of this model for the active learning experiments.
>
> > Line 21: I feel like have read the first sentence many times, almost every year at NeurIPS, in papers on computational neuroscience…
>
> We thank the reviewer for pointing this out and will revise for the final version accordingly.
>
> > The exposition in Section 4 is overly technical. It would help if the high level narrative of the paper and the essential steps where explained on a somewhat more intuitive level.
>
> We apologize for any confusion this section may have caused and will add additional intuition and explanation to the final version using the provided extra page.
>
> Intuitively, the chief contention of Section 3 is that the neural population data can be effectively modeled by low-rank dynamics. Section 4 then seeks to determine how measurements should be taken so that, given this low-rank structure, the estimation error of the population dynamics may be minimized as effectively as possible. The results of Section 4.1 provide a quantification of the estimation error of the nuclear norm estimator for such low-rank models. Section 4.2 then proposes an algorithm which chooses measurements that minimize the estimation error, as quantified in Section 4.1. Together, then, these results show how to choose measurements to estimate the neural population dynamics as effectively as possible, assuming they exhibit low-rank structure.
>
> > No errorbars in Figure 3 and 4.
>
> Figures 3 and 4 do include error bars denoting 1 standard error, they are simply too small to be visible in most plots, as the variance from trial to trial we observed was quite small. See, for example, the blue curve in the “Best” row of Figure 4(b) for a plot where the error bars are visible.
>
> > Line 138: Why did the authors choose the term Causal connectivity matrix? I don’t understand why H would be a causal estimate. It would be good if the authors could elaborate or choose another term.
>
> We use the term “causal connectivity matrix” because that matrix summarizes the causal response of each i-th neuron to photostimulation of each j-th neuron. We use the term “causal” to stress that these relationships are learned from direct causal perturbations to the neural population. By contrast, the vast majority of published work on neural population dynamics involves fitting dynamical models to passively obtained neural data. Due to the lack of causal manipulations in those studies, one cannot distinguish whether statistical relationships arise between neurons due to correlation (e.g., due to a shared upstream influence) versus causation (e.g., neuron i directly influences neuron j) . Such correlative relationships are typically referred to as “functional connectivity,” and we intended for our “causal connectivity” to convey the additional causal interpretability afforded. We will update our paper to clarify this distinction.
>
> > Wouldn’t theorem 2 be sufficient or the theorem from ref 67 for the point the authors want to make? How does Theorem 2 reconcile with the claim of the authors in line 176 that this is the first such bound.
>
> Theorem 2 is a lower bound—that is, it shows how much estimation error any estimator must incur, i.e. it is not possible to achieve estimation error less than that given in Theorem 2. Theorem 1 is an upper bound—it bounds the estimation error achieved by the particular estimator we are considering here (the nuclear norm estimator). The lower bound is not novel, but the upper bound we provide is.
>
> For any given estimator, it is important to understand how the measurements taken affect the estimation error for that particular estimator, so that measurements can be taken to minimize the estimation error for that estimator. The lower bound of Theorem 2 does not provide this insight, as it holds for any estimator; Theorem 1 does quantify how the estimation error scales for the nuclear norm estimator, and therefore motivates our choice of sampling.
>
> > Section 4.2: I understand that Algorithm 1 somehow falls out of the theory, but it would be helpful for me if the authors could convey the intuition between $\lambda^V$ and $\lambda^{uniform}$, and why one wants to mix them.
>
> By combining Theorem 1 and equation (4.3), our results show that the estimation error scales with a combination of two terms: the first corresponding to the input power in the directions spanned by the low-rank subspace, and the second corresponding to the span of the inputs in all directions. This suggests that, to minimize estimation error, we should devote some amount of input energy to the directions spanned by the low-dimensional subspace, and some amount to cover every direction. This is precisely what Algorithm 1 instantiates: $\lambda^V$ chooses the inputs to target the low-dimensional subspace, while $\lambda^{uniform}$. By mixing these allocations we ensure that we play inputs targeting both relevant objectives.
>
> > Line 205: What is meant by “played isotropically”? Algorithm 1: What would be a typical range for l (log T)? Is it possible that the 2^l loop in step 5 becomes infeasibly long?
>
> By “isotropic”, we simply mean the amount played in every possible direction. In Algorithm 1, T can be chosen by the user based on how many samples they can collect, so, for example, the real data we have contains approximately 2000 input-response segments, so this would correspond to $T = 2000$. As such, $2^{\ell}$ will only be as large as the user desires, so it will not become infeasibly large.
>
> > Line 248: Is this the same T as in Algorithm 1?
>
> Yes.

---

### Official Review · Reviewer_cE26 · 2024-07-12

**Soundness:** 3
**Presentation:** 3
**Contribution:** 3
**Rating:** 7
**Confidence:** 2

**Summary:**

The advent of holographic optogenetics has brought about an unprecedented level of specificity in the way we stimulate and measure the activity across the neural population. The authors propose methods for efficiently determining effective photostimulation patterns to study neural population dynamics using two-photon holographic optogenetics. They propose a novel active learning approach that leverages a low-rank linear dynamical systems model to determine patterns of stimulation with the highest information capacity aiming to characterize the neural population dynamics by minimizing the necessity for data collection.

**Strengths:**

The authors use holographic 2-photon calcium imaging in order to optimize the pattern of photo-stimulation, something which will be useful for many subsequent studies and experiments.
The way in which the authors recover low-rank matrices with the novel use of the nuclear norm and also bound those low rank matrices seems to work well for linearly approachable problems and can also be applied further to other scientific domains.
Furthermore, they explain the reduced autoregressive model fairly well and provide proofs for 2 new theorems that will ultimately bound the low rank matrices.
Studying the aforementioned low rank matrices through the lens of active learning gives a novel perspective on how to optimize data acquisition and handling. The paper also makes clear how active learning is used to effectively opt for those stimulations that accelerate the dynamics estimation.
It appears that this method can significantly reduce the data required for accurate estimation, which is important given the time-limits that these experiments enforce.

**Weaknesses:**

The dataset is very simple and does not contain anything to enrich the dynamics of the neural population except for random noise. There is not a task in which the animal is involved, at least as presented in the text. The modality of choice is movement which has fairly straightened out dynamics and well characterized, low dimensional neural trajectories. Will the approach work the same in a sensory modality? This could significantly cut off from the model’s wide applicability.

If the model is used in a more complex modality, the structure of the causal connectivity matrix should be significantly altered, thus current assumptions might not hold true.

Some of the figures are not adequately explained in their legends and some of the ranks are obsolete (Figure 3 and Figure 4)

**Questions:**

Part of Sector 4.1 seems more like a methods sector. The authors could try to make the algorithm explanation a bit more intuitive and justify a bit more the rationale behind each equation.

The authors could potentially expand their repertoire of datasets to different modalities and more complex behavioral paradigms as this will help them infer whether the algorithm can be effective in more realistic datasets. It would be interesting to try different causal connectivity matrices with more complex data. I understand though that such datasets might not be available.

**Limitations:**

The authors provide 3 limitations of their approach, namely the uniformity in which the causal connectivity matrix is being approached across all neurons, the applicability of their approach to non-linear dynamics and whether their algorithm can be used online during the experiment.

 I would also add as a limitation the very specific and constrained real-dataset in which the algorithm has been tested on and whether it can have a generally good performance in other, more complex frameworks. For example, the neural activity structure of sensory areas could differ significantly from that of motor areas.

Finally, they discuss that the don’t have a clear validation test with new recordings in which the optimal stimulation patterns could be utilized.

In any case, the direction of this paper is very important for future experiments.

---

> ### Author Rebuttal · Authors · 2024-08-07
>
> > The dataset is very simple and does not contain anything to enrich the dynamics of the neural population except for random noise…
>
> > The authors could potentially expand their repertoire of datasets to different modalities and more complex behavioral paradigms as this will help them infer whether the algorithm can be effective in more realistic datasets…
>
> > I would also add as a limitation the very specific and constrained real-dataset in which the algorithm has been tested on and whether it can have a generally good performance in other, more complex frameworks…
>
> We agree that it would be interesting to further characterize our dynamical models and active learning approaches across a variety of brain areas (e.g., including sensory areas) and while the animal engages in behaviors that may lead to different neural population dynamics. Our collaboration and many others around the world are particularly invested in understanding population dynamics in the motor cortex, however, and hence we have focussed this paper on motor cortical data. Nonetheless, we do hope to extend our techniques to other brain areas in future work. In that vein, we will update the limitations discussed in the paper to assert that further experiments will be required to assess how well our approach generalizes to other brain areas and behavioral paradigms. We found that low-rank autoregressive linear dynamics were sufficiently expressive to model the datasets described in the paper and did not require the added expressivity of the nonlinear dynamical models we fit (see Fig 2, “GRU”). In future work, we hope to repeat these analyses on data collected while the animal engages in a behavioral task. If these data demand nonlinear dynamical models, that could justify developing active learning approaches in those nonlinear models.
>
> > Some of the figures are not adequately explained in their legends and some of the ranks are obsolete (Figure 3 and Figure 4)
>
> We thank the reviewer for pointing this out and will add a more detailed description to the legend of each figure for the final version, as well as working to make the figures more concise.
>
> > Part of Sector 4.1 seems more like a methods section. The authors could try to make the algorithm explanation a bit more intuitive and justify a bit more the rationale behind each equation.
>
> We thank the reviewer for pointing this out and will add additional intuition and explanation to the final version using the extra page of space. Please see also our response to Reviewer d6y7 for additional explanation of Section 4.

---

> > ### Comment · Reviewer_cE26 · 2024-08-09
> >
> > I would like to thank the authors for the response to my comments and the clarifications that will appear in the manuscript.

---

### Official Review · Reviewer_YQWs · 2024-07-13

**Soundness:** 3
**Presentation:** 3
**Contribution:** 3
**Rating:** 6
**Confidence:** 3

**Summary:**

The paper proposes an active learning framework for choosing the next set of neurons to stimulate to best inform a dynamical model of the neural population activity. The active learning procedure takes advantage of the low-rank structure of the dynamical systems model. With synthetic and real datasets, the authors demonstrate that the approach can obtain as much as a 2x reduction in the amount of data required to reach a given predictive power.

**Strengths:**

- Clarity: The paper is well-written, with appropriate equations and figures. Sections 3 and 4 especially present complex mathematical derivations and theorems with details, allowing readers with little expertise to follow the paper.

- Significance: Due to the enormous space of potential stimulations and the time-consuming nature of neuroscience experiments, a reduction in the number of experiments via active learning is crucial for fast advancement in neuroscience.

**Weaknesses:**

- Experimental analysis: Although the paper has interesting experimental results, I would like further explanations of the results. For example, what is causing the discrepancy between the best and worst cases in Figure 4?

- Comparisons with other methods: The authors cite multiple papers on actively designing inputs for system identification. Another recent paper that could be relevant is [1]. However, the paper compares the proposed method with too simple baselines, such as random or uniform stimulation. Comparisons with other methods for active design of experiments would be crucial to demonstrate the effectiveness of the method proposed by the authors.

[1] Jha, A., Ashwood, Z. C., & Pillow, J. W. (2024). Active Learning for Discrete Latent Variable Models. Neural computation, 36(3), 437-474.

**Questions:**

- Minor comment:
The authors could cite more recent uses of data-driven dynamical models for neural data [1, 2].

[1] Karniol-Tambour, O., Zoltowski, D. M., Diamanti, E. M., Pinto, L., Tank, D. W., Brody, C. D., & Pillow, J. W. (2022). Modeling communication and switching nonlinear dynamics in multi-region neural activity. bioRxiv, 2022-09.

[2] Lee, H. D., Warrington, A., Glaser, J., & Linderman, S. (2023). Switching autoregressive low-rank tensor models. Advances in Neural Information Processing Systems, 36, 57976-58010.

In particular, the authors could consider replacing the low-rank AR models with the model from [2] when fitting models with more lags. While low-rank AR models can be overparameterized easily when incorporating more lags, the model from [2] has fewer parameters than standard low-rank AR models and draws a connection to LDSs.

- How is the lag hyperparameter chosen for the AR models?

- In Figures 3 and 4, it's hard to know whether certain estimation error is good enough. Some visualizations of the learned matrices or drawing horizontal lines in the graph to represent the "best" case scenario (e.g., parameters fitted to an extremely large set of input/observation pairs) would be helpful.

**Limitations:**

- As the authors noted, one limitation could be that the low-rank AR-k models may not be effective enough to capture potential nonlinear neural dynamics. Another limitation is that the experiments of the paper are offline. It will be important to test the method in real-time during closed-loop experiments.

---

> ### Author Rebuttal · Authors · 2024-08-07
>
> > Experimental analysis: Although the paper has interesting experimental results, I would like further explanations of the results. For example, what is causing the discrepancy between the best and worst cases in Figure 4?
>
> In Figure 4, the “Best” and “Worst” plots are the best performing and worst performing train-test split, respectively. It is not entirely clear why our method performs better on some train-test splits than others, as compared with passive approaches, where the train-test split is chosen based on the unique input patterns in the dataset. For some train-test splits there may be a stronger correlation between inputs in the train and test set than for others, or the particular train-test split may make it harder or easier to learn the relevant low-rank structure, both of which could affect the performance of active learning vs passive learning. More investigation into which settings active learning provides a substantial gain is an interesting direction for future work.
>
> We are happy to provide additional explanation on particular points, if the reviewer could bring these to our attention. We will also use the extra page in the camera-ready version to add further explanation to the paper.
>
> > Comparisons with other methods: The authors cite multiple papers on actively designing inputs for system identification. Another recent paper that could be relevant is [1]. However, the paper compares the proposed method with too simple baselines, such as random or uniform stimulation. Comparisons with other methods for active design of experiments would be crucial to demonstrate the effectiveness of the method proposed by the authors.
>
> We make several points regarding the chosen baselines. First, for the model class we consider, i.e. AR-k linear models, the approaches we test against are the standard approaches—a fixed design where measurements are chosen to cover all directions is the standard choice for linear settings (see e.g. [Pukelsheim, 2006]), and randomly choosing inputs is also a common benchmark [Simchowitz et al., 2018]. We are not aware of commonly used approaches that differ significantly from these approaches in linear settings. Furthermore, we are not aware of any other approaches for choosing photostimulation patterns in a targeted manner that we could compare against. We hope our work motivates further research into this question, and expands the set of baselines relevant to such settings.
>
> Second, our proposed approach is essentially Fisher Information maximization for low-rank AR-k linear models, while the approach proposed in [1] is a Bayesian instantiation of Fisher Information maximization for a particular class of latent-variable models (see e.g. [Chaloner & Verdinelli, 1995] for justification of this). Therefore, our approach relies on the same principle as that proposed in [1]—Fisher Information maximization—simply for a different model class (low-rank AR-k linear models vs latent variable models). Whether or not the exact approach of [1] is relevant in our setting is therefore primarily a question of the chosen model class. As our results in Section 3 illustrate, the data are effectively fit by a low-rank linear model, and it does not appear that latent variable models (which our GRU model can be seen as an example of) yield a significant improvement. Furthermore, [1] highlights several shortcomings of their proposed approach which would make it difficult to apply in our setting: they state that applying their approach to high-dimensional outputs is difficult (we consider output dimension in the range 500-600), the input design only allows selecting inputs from a discrete set of candidates (while we require continuous inputs), and their approach is not applicable when the state-transition depends on the inputs (which is the case in our setting). Given this, we do not believe that the exact method of [1] is particularly relevant in our setting.
>
> If the reviewer has additional benchmarks that would be appropriate to consider with that we have missed, we would be happy to consider.
>
> Pukelsheim, Friedrich. Optimal design of experiments. Society for Industrial and Applied Mathematics, 2006.
>
> Simchowitz, Max, et al. "Learning without mixing: Towards a sharp analysis of linear system identification." Conference On Learning Theory. PMLR, 2018.
>
> Chaloner, Kathryn, and Isabella Verdinelli. "Bayesian experimental design: A review." Statistical science (1995): 273-304.
>
> > Minor comment: The authors could cite more recent uses of data-driven dynamical models for neural data…
>
> We thank the reviewer for noting these relevant references, and we will certainly  incorporate them into the final version of the paper. We appreciate the suggestion of using the model from [2], and hope to investigate this model in future work.
>
> > How is the lag hyperparameter chosen for the AR models?
>
> We chose this via standard cross-validation procedures: we fit models to the data with several different lag parameters, and then evaluated their performance on held-out data. In practice we found that increasing the lag parameter past 3 or 4 did not substantially improve the model’s fitting ability, so all results in the paper are with a lag parameter of $k = 4$. We will add details on this to the final version of the paper.
>
> > In Figures 3 and 4, it's hard to know whether certain estimation error is good enough. Some visualizations of the learned matrices or drawing horizontal lines in the graph to represent the "best" case scenario (e.g., parameters fitted to an extremely large set of input/observation pairs) would be helpful.
>
> Please see the rebuttal pdf for visualizations of the performance for different error measures. In particular, we illustrate the causal connectivity matrix for different levels of the estimation error given in Figure 3, and the neural time-series behavior for different levels of the MSE given in Figure 4.

---

> > ### Comment · Reviewer_YQWs · 2024-08-09
> >
> > I would like to thank the authors for their response to my comments and the clarifications and modifications that will appear in the manuscript. I would like to increase the score from 5 (Borderline accept) to 6 (Weak accept).

---

### Author Rebuttal · Authors · 2024-08-07

We thank each of the reviewers for their helpful feedback, and will work to incorporate all suggestions in the final version. We have addressed specific questions in the following, and are also attaching a pdf with additional visualizations, as requested in some of the reviews.

---

### Decision · Program_Chairs · 2024-09-25

**Decision:**

Accept (poster)

**Comment:**

This paper describes novel active learning methods for optogenetic perturbation experiments aimed at reducing the amount of data required to infer accurate models of neural dynamics.  All three reviewers felt that it was above the threshold for acceptance, and I'm pleased to report that it has been accepted to NeurIPS.  Congratulations!  Please revise the manuscript according to the reviewer comments and discussion points.